# Numerical Investigation of Halbach-Array-Based Flexible Magnetic Sensors for Wide-Range Deformation Detection

**DOI:** 10.3390/s25237240

**Published:** 2025-11-27

**Authors:** Yina Han, Shuaiqi Zhang, Chenglin Wen, Jie Han, Wenbin Kang, Zhiqiang Zheng

**Affiliations:** 1Ministry of Education Key Laboratory for Non-Equilibrium Synthesis and Modulation of Condensed Matter, Shaanxi Province Key Laboratory of Advanced Functional Materials and Mesoscopic Physics, School of Physics, Xi’an Jiaotong University, Xi’an 710000, China; 3297895834@stu.xjtu.edu.cn (Y.H.); xjtuzsq@stu.xjtu.edu.cn (S.Z.); 1311878470@stu.xjtu.edu.cn (C.W.); 2Department of Mechanical Engineering, City University of Hong Kong, Hong Kong 999077, China; wenbin.kang@cityu.edu.hk; 3Department of Biomedical Engineering, City University of Hong Kong, Hong Kong 999077, China

**Keywords:** tactile sensing, magnetic sensor, Halbach array, embodied intelligence, numerical investigation

## Abstract

**Highlights:**

A Halbach-array-based magnetic tactile sensor is proposed, where the soft magnetic layer is decoupled from the rigid Hall sensing unit to achieve wide-range, high-sensitivity and remote strain detection. Finite element simulations show that the Halbach configuration strengthens and homogenizes the magnetic field while reducing its spatial dimensionality, enabling simplified data processing, faster response, and scalable design for next-generation flexible tactile sensing in wearable and robotic applications.

**What are the main findings?**
Decoupled architecture for enhanced magnetic coupling. Separating the flexible PDMS-based Halbach magnetic array from the fixed Hall sensor preserves field uniformity under large deformation and enables mT-level detection at a sensing distance up to 15 mm.Dimensionality reduction of magnetic field distribution. The Halbach configuration transforms a complex 3D magnetic field into a near 1D directional field, enhancing sensitivity, simplifying signal processing, and supporting scalable high-resolution tactile sensing.

**What are the implication of the main findings?**
The decoupled Halbach-array design provides a general strategy for achieving wide-range, high-sensitivity magnetic sensing in flexible systems, enabling more reliable tactile perception in wearable and robotic applications.The dimensionality reduction of magnetic fields offers a new approach to simplify signal processing and enhance real-time sensing speed in future intelligent tactile interfaces.

**Abstract:**

Flexible magnetic tactile sensors hold great promise for wearable electronics and intelligent robotics but often suffer from limited strain range and complex magnetic field variations due to rigid-soft coupling between the Hall sensor and magnetic layer. In this study, we propose a Halbach-array-based magnetic tactile sensor that structurally decouples the soft magnetic deformation layer from the rigid Hall sensing unit. The sensor embeds k = 2 Halbach-configured magnetic cubes within a PDMS matrix, while the Hall element is fixed at a remote, rigid location. Numerical analysis using COMSOL Multiphysics demonstrates that the Halbach configuration enhances magnetic field strength and uniformity, achieving mT-level detection even at a distance of 15 mm. Moreover, the Halbach array effectively reduces the field distribution from three-dimensional to one-dimensional, enabling stronger directionality, simplified data processing, and higher sensing frequency. This work establishes a theoretical framework for wide-range, high-precision magnetic tactile sensing through magnetic field tailoring, providing valuable guidance for the design of next-generation flexible sensors for wearable, robotic, and embodied intelligence applications.

## 1. Introduction

Tactile sensing plays an essential role in nature, enabling organisms to perceive their surroundings, respond to external stimuli, and manipulate objects with remarkable dexterity. Human skin, in particular, exhibits extraordinary capabilities in detecting pressure, texture, temperature, and vibration, forming the foundation of human-object interaction and environmental awareness [1]. Inspired by such biological systems, the development of electronic skin (E-skin) has become a central pursuit in robotics and wearable technology, aiming to replicate the multimodal tactile perception of natural skin for human-like robots and intelligent systems [2,3,4,5], with many new possibilities for flexible electronics in biomedical applications [6,7,8]. This tactile functionality is critical not only for precision manipulation and safe physical human–robot interaction but also for achieving embodied intelligence, where a robot’s perception and motion are tightly coupled through its physical body and sensory feedback [9,10,11,12].

Over the past decade, significant advances have been made in tactile sensing technologies for flexible and stretchable E-skins. Various sensing mechanisms, such as resistive, capacitive, piezoelectric, optical, and barometric transduction, have been explored to measure contact forces, surface deformation, and vibration [13,14,15,16]. Resistive sensors are widely used due to their simple fabrication and signal readout, but they often suffer from hysteresis and limited stability under repeated strain [17,18,19,20,21]. Capacitive sensors offer high sensitivity and fast response, yet they are easily influenced by environmental factors such as humidity and electromagnetic noise [22,23,24,25]. Piezoelectric and triboelectric sensors provide self-powered operation and high dynamic response, but they typically measure only transient signals and are difficult to integrate over large areas [26,27,28,29]. Optical and barometric sensors exhibit good linearity and can achieve distributed sensing, though their structures are often complex, bulky, or difficult to miniaturize [30,31,32,33,34].

To overcome these limitations, magnetic-based tactile sensing has emerged as a promising approach due to its unique combination of mechanical flexibility, environmental robustness, and high spatial resolution [35,36,37,38]. This is a broad field encompassing diverse strategies. Some approaches utilized the magneto-elastic effect, where applied stress directly modulates the magnetic properties of a flexible film (e.g., FeNi) [39]. Other works have developed flexible sensors based on magnetoimpedance (MI) structures that are highly sensitive to deformation [40]. A third, widely adopted strategy, which forms the basis for our work, involves a soft composite layer (a soft matrix embedded with micro- or nano-magnets) that mechanically deforms over a fixed Hall or magnetoresistive sensing element. In this configuration, deformation induces displacement or rotation of the magnetic source, causing measurable magnetic field variations [41,42,43]. This composite-based Hall-effect approach, in particular, offers several key advantages: (1) Easy deployment and structural simplicity, since magnetic coupling allows non-contact signal transmission without complex wiring or optical paths; (2) Magnetic-electrical decoupling, which enables the sensitive electronics (e.g., Hall elements) to be isolated from mechanical strain, thereby improving device durability and long-term stability; (3) High spatial resolution, achievable by designing magnetic arrays or encoding field gradients for super-resolution tactile mapping; and (4) Compatibility with soft materials, enabling conformal integration onto curved or deformable surfaces [44,45,46].

Despite these advantages, magnetic tactile sensors still face several key challenges that limit their performance in practical applications. First, as noted in designs by Yan et al. [47], when the Hall sensor is directly attached behind the magnetic film, the rigid-soft interface restricts the mechanical deformability, leading to a nonlinear response under large strain. To address this, “split-type” or decoupled architectures have been proposed, such as by Dai et al. [35] and Sundaram et al. [48], which physically separates the deformable magnetic layer from the rigid sensor. These designs improve robustness and enable remote sensing. Second, the magnetic field from conventional magnets is non-uniform and three-dimensional, complicating calibration. While soft magnetic composites allow for novel field profiling (e.g., sinusoidal or centripetal) to achieve force decoupling, the resulting 3D fields often require complex analytical models or machine learning for interpretation. Third, the magnetic signal intensity rapidly decays with distance, which constrains the sensing range and makes it difficult to achieve accurate measurements when the magnetic and electronic units are separated. These issues hinder the realization of high-precision, wide-range magnetic tactile sensors that can operate reliably under large deformation, an essential requirement for wearable and robotic E-skins [38,47,49,50,51].

To address these challenges, this study proposes a Halbach-array-based magnetic tactile sensor in which a soft PDMS magnetic layer, embedded with k = 2 Halbach-configured magnetic cubes, is structurally decoupled from a rigid, remotely positioned Hall sensing unit. A comprehensive numerical framework is developed using COMSOL Multiphysics to investigate the magnetic field distribution, uniformity, and strength as functions of key design parameters. The aim is to establish design principles that enable wide-range, high-precision, and mechanically flexible magnetic sensing.

## 2. Proposed Flexible Magnetic Tactile Sensor

### 2.1. Halbach-Array-Based Magnetic Sensor Design and Working Principle

The proposed flexible magnetic tactile sensor is composed of two primary components: a magnetic film and a Hall sensing unit. Unlike conventional magnetic tactile sensors, which typically employ pre-magnetized composites consisting of dispersed magnetic microparticles within an elastomer matrix, our design utilizes discrete millimeter-scale permanent magnets embedded directly into a pure elastomer substrate. These magnets are arranged in a specific orientation to form a Halbach array operating in the k = 2 configuration, as illustrated in Figure 1a. The Hall sensor is positioned beneath the central region of the Halbach array at a predetermined distance to detect the resulting magnetic field distribution. This work mainly focused on the magnetic field generation section, and our further work will introduce the experimental results with data collected from the 3D Hall sensor.

In a Halbach array, two key parameters define the magnetic configuration: the revolution angle (α) and the rotation angle of each magnet (β). The relationship between these parameters is expressed as k = β/α. When k = 2, the array exhibits a unique magnetic property, that its internal magnetic field is reinforced and highly directional, while the external field on the opposite side is significantly weakened. At the center of the array, this configuration effectively modulates the originally three-dimensional magnetic field into a quasi-one-dimensional distribution, producing a uniform and concentrated field along a single direction.

This one-dimensional magnetic field characteristic forms the fundamental principle of the present design. It allows the magnetic film to maintain strong and consistent field strength even under deformation, thereby simplifying field analysis and enhancing the precision and stability of tactile signal detection, as schematically shown in Figure 1b.

### 2.2. Finite Element Method for Sensor Modeling

Numerical simulations were conducted using COMSOL Multiphysics 6.2, employing the Magnetic Fields, No Currents module in three-dimensional mode to analyze the magnetic field behavior of the proposed sensor. The individual magnets were modeled as cylindrical permanent magnets with a diameter of 2 mm and a height of 2 mm. The overall Halbach array diameter varied between 10 mm and 30 mm; unless otherwise specified, all simulation results presented in this work are based on an array with a diameter (*D*) of 10 mm, which was experimentally optimized by considering the trade-off between magnetic film deformation and sensor array density.

As illustrated in Figure 2a, the model domain consisted of a confined air region with a diameter of 60 mm. The central analysis zone was defined as a cylindrical region with a diameter of (*D* = 4) mm and a height of 20 mm, extending from +5 mm to −15 mm along the z-axis. The magnetic material was defined as sintered NdFeB (N52), characterized by a recoil permeability of 1.05, a remanent flux density of 1.44 T. The baseline design parameters were chosen to be representative of tactile sensing applications, balancing performance with practical implementation. The individual magnets (sintered NdFeB N52, 2 mm diameter, 2 mm height) were selected as they provide the highest remanent flux density (~1.44 T) in a compact, commercially available form factor suitable for embedding in a soft substrate. The array diameter (*D*) range of 10–30 mm was chosen to match the typical scale of a single tactile pixel for robotic fingertips and e-skin, which often required magnetic unit ranging from approximately 8 mm to 20 mm in diameter or width [35,36,37,38,39,40,41,42,43,44,45,46,47]. This range also represents a practical trade-off between spatial resolution, which our analysis in Section 3.4 shows flux density is enhanced at smaller diameters, and the physical challenges of precisely assembling the array. The remote sensing distance (analyzed up to z = −15 mm) was a key parameter of this investigation. A primary objective was to verify that the Halbach array could maintain a strong, mT-level signal at this significant distance. This contrasts with many soft-composite-based sensors, which often drop to the microtesla (μT) level at similar distance, resulting in a much lower signal-to-noise ratio (SNR).

A fine mesh was applied to ensure numerical accuracy, with element sizes of 0.2 mm in the central region, 0.3 mm in the magnets, and 0.32–2.56 mm in the surrounding air. Free tetrahedral elements were used throughout. To ensure the “degree of instability” of the solution was minimized, a mesh-convergence study was performed; the mesh was refined until the change in peak magnetic flux density was below 0.1%, confirming numerical stability and independence from mesh discretization. To isolate the magnetic response induced purely by magnet rotation, the magnets were constrained to rotate in-plane about their central axes (*θ* up to 60°) without translation. This assumption reflects the dominant deformation mode in the soft magnetic film, where external loading primarily induces small-angle rotations rather than significant translational motion.

While this model captures the essential rotation-driven modulation of the magnetic field, the main mechanism governing sensor sensitivity, it does not include out-of-plane shifts or magnet–matrix interactions that may arise in real deformation. These coupled magneto-mechanical effects will be incorporated in future studies. Figure 2c shows the simulated x–z plane field distribution as *θ* increases from 0° to 30°. The field remains strongly aligned along the x-axis with only a minor z-component, and its orientation remains stable during rotation, confirming the directional robustness of the Halbach configuration.

## 3. Results

### 3.1. Magnetic Field Profile in the Central X–Z Plane During Rotation of the Halbach Array

To examine the directional characteristics of the generated magnetic field, the magnetic field distribution was analyzed in the central x–z plane (where y = 0 mm). Within the analyzed region, seven horizontal sampling lines and one central vertical line were defined to evaluate the field directionality and uniformity, as illustrated in Figure 3.

Along the central vertical line (x = y = 0, z ranging from +5 mm to −15 mm), the magnetic field vectors were found to be almost perfectly aligned with the x-axis, while the flux components along the other axes were effectively suppressed. This directional uniformity indicates that the Halbach configuration successfully reduces magnetic field complexity and minimizes interference, which in turn simplifies data acquisition and significantly decreases computational load during signal processing.

For the horizontal lines at various z-coordinates, the magnetic field within the central region of the array exhibited strong uniformity and minimal z-axis components. Near the periphery of the analysis region, however, the z-component increased due to the influence of localized three-dimensional magnetic fields around the permanent magnets. As the observation point moved farther from the magnet plane, the z-component gradually diminished, reflecting the attenuation of vertical field influence with distance.

Importantly, during the simulated rotation of the magnets within the Halbach array, the directional profile of the generated magnetic field remained consistent across different rotation angles. This stability implies that variations in the magnetic signal are primarily governed by changes in field strength rather than direction, thereby facilitating more straightforward data interpretation and calibration.

It should be noted that the arrow lines shown in Figure 3 indicate only the direction of the magnetic field vectors at corresponding points, with uniform arrow lengths for visualization clarity; they do not represent the actual magnitude of the magnetic field.

### 3.2. Magnetic Field Profile in X–Y Plane During Rotation of Halbach Array Units

In addition to the x–z plane, the field distribution in the plane of the magnets themselves (z = 0 mm) was examined, as shown in Figure 4. This analysis is important for understanding the stability of the magnetic field source, not the sensing response. The actual sensing region is located remotely in the z < 0 space. The maximum magnetic flux density occurred near the edges adjacent to the magnets, decreasing from 169 mT to 140 mT as the Halbach array rotated from 0° to 50°. Conversely, the minimum values appeared at the boundaries between neighboring magnets, regions that should be carefully considered and avoided in experimental measurements.

In the central region, the magnetic field strength remained relatively uniform, and no significant spatial pattern changes were observed during magnet rotation, even when the rotation angle *θ* reached 50°. Moreover, for regions where *D* < 4 mm, the magnitude change in magnetic flux density is less than 0.5 mT. These results demonstrate the stability of the Halbach array’s magnetic field configuration and suggest that the proposed flexible sensor possesses strong resilience against external magnetic interference.

Figure 5 shows the simulated magnetic flux density in horizontal cross-sections of the Halbach array at different probe heights (z) and rotation angles (*θ*). Rows correspond to axial positions (z = 5 mm, −5 mm, −10 mm, −15 mm) and columns to rotation angles (*θ* = 0°–30°). Because the sensor is positioned below the array (negative z), the top row (z = 5 mm) represents the suppressed field side, while the lower rows illustrate the enhanced sensing region.

The simulations reveal a clear rotation-induced asymmetry. At *θ* = 0°, the field is symmetric about z = 0, with identical distributions at z = +5 mm and −5 mm. Once rotation begins, this symmetry is lost: the Halbach configuration strengthens the field in the sensing region (z < 0) and weakens it above the array (z > 0). With increasing *θ*, the field evolves nonlinearly despite the axial peak flux varying nearly linearly, highlighting spatial complexity central to the sensor’s modulation behavior.

For a fixed probe height, increasing *θ* causes the field pattern to rotate synchronously, with high- and low-field regions shifting around the central axis. This strong coupling between magnet orientation and field direction allows the global field both to be tuned and to encode the array’s rotational state.

Overall, probe height primarily affects the magnitude and uniformity of the flux density, while rotation angle sets its directional features. The consistent field pattern during rotation suggests robust immunity to external magnetic disturbances.

### 3.3. Magnetic Field Profile Along Central Z-Axis During Rotation of Halbach Array Units

To characterize the axial evolution of the field, the magnetic flux components along the central z-axis were evaluated as functions of *θ* (Figure 6). The B_x_ component (Figure 6a) remains in the mT range and dominates the response, while B_y_ and B_z_ stay at μT levels, confirming strong one-dimensional field confinement with minimal crosstalk.

As summarized in Figure 6b, the B_x_ peak shifts linearly toward negative z with a slope of −0.047 mm/° (R^2^ = 0.997), and its amplitude decreases linearly with a slope of −0.021 mT/° (R^2^ = 0.990). These highly linear trends demonstrate a stable and predictable rotational response. Overall, the Halbach array converts the 3D field into a robust 1D profile with mT-level strength even at a 15 mm stand-off, providing a clean, calibration-friendly signal ideal for directionally selective tactile sensing.

Figure 7 presents the magnetic flux density along the X-axis at several Z-heights within the Halbach array. Across all positions, the field is dominated by the B_x_ component and B_y_ and B_z_ remain strongly suppressed, while B_y_ shows only μT-level oscillations (Table 1). As the difference in these two directions is on the order of 10^4^–10^7^, this confirms excellent rejection of lateral (y-axis) disturbances and minimal crosstalk due to misalignment.

The B_x_ profiles show smooth, predictable, and nearly linear changes with rotation angle *θ*, forming a clean basis for rotation or shear sensing. As Z decreases toward the magnets, the curves become sharper and exhibit a larger dynamic range, while at greater distances (e.g., z = −15 mm) the profiles flatten but maintain strong linearity.

The B_z_ component exhibits more complex behavior. Close to the magnets (e.g., z = −3 mm), B_z_ shows a pronounced polarity flip at x = 0 and a distinctly nonlinear shape. With increasing distance, this pattern gradually transitions into a more monotonic, quasi-linear form. This evolution means that B_z_ encodes additional spatial information: at small Z, the nonlinear signature could allow simultaneous estimation of rotation and axial position using data-driven models, whereas at large Z the response becomes simpler but loses Z-specific detail. Results in Figure 7 demonstrate that B_x_ provides a clean and highly linear rotational signal, B_z_ carries depth-dependent spatial cues, and B_y_ remains negligible, together offering a robust magnetic field environment for stable and decoupled sensing.

A key claim of this work is the high sensitivity enabled by the Halbach configuration, expressed through its strong SNR. As shown in Figure 6a, the array produces mT-level signals even at 15 mm, 2–3 orders stronger than the μT-level fields typical of soft magnetic composites [35,50]. This large signal ensures a high SNR and a clean, linear response, as seen in Figure 6b.

To quantitatively assess the variation in magnetic field strength with distance and rotation angle, the total magnetic flux density (B) and its X-directional component (B_x_) were evaluated at the center of the Halbach array (X = 0 mm, Y = 0 mm), as shown in Figure 8. The results reveal that, above the array (Z ≥ 0 mm), the magnetic field intensity decreases with increasing rotation angle, while below the array, the field strength increases with distance and exhibits enhanced linearity at larger |Z| values. This behavior indicates effective confinement of the magnetic field near the lower surface, which amplifies the detectable magnetic signal and improves the sensor’s signal-to-noise ratio (SNR) performance. Additionally, the close agreement between B and B_x_ confirms that the Y- and Z-directional components are effectively suppressed within the central region by four to five orders of magnitude. These quantitative insights provide a solid theoretical basis for optimizing sensor placement and refining data processing strategies.

### 3.4. Magnetic Field Profile When Adjusting Halbach Array Size

To systematically investigate the influence of array diameter on magnetic field modulation, a series of simulations was conducted. The magnetic flux distribution of six magnet-consistent Halbach arrays with pattern diameters of 10 mm, 20 mm, and 30 mm was analyzed. Similarly to previous analysis, each magnet was modeled as a 2 × 2 cylindrical N52 element.

Figure 9a–c present the B_x_ distributions in the x–y plane at Z = −6 mm within the array. This analysis reveals a critical design trade-off. The smaller-diameter array (10 mm) generates stronger fields and, more importantly, more pronounced spatial variation (i.e., a steeper dBx/dx gradient). This implies that a smaller array diameter provides higher spatial resolution and greater sensitivity to positional changes. Conversely, the 30 mm array exhibits a non-monotonic, “M-shaped” profile, with two distinct edge peaks. This is highly undesirable for sensing, as a single B_x_ value could correspond to two different x-positions, creating signal ambiguity.

Figure 9d–h shows the magnetic field distributions along the Z-axis. While the overall field magnitude decreases with increasing array diameter, the fundamental spatial distribution pattern and linear response to rotation remain consistent. This result is significant as it demonstrates the generality and scalability of the Halbach design principle.

Furthermore, a parameter uncertainty analysis was applied to the 10 mm array, as shown in Figure 9g. The shaded bands (representing a ±3% variation in remanence, B_r_) indicate that while the absolute peak intensity scales with magnet strength, the peak position and the highly linear response to rotation remain stable, confirming the robustness of the design. Based on these findings, minimizing the array diameter (e.g., 10 mm or 20 mm) is recommended to maximize spatial resolution and sensitivity while avoiding the signal ambiguity observed in larger arrays.

## 4. Discussion

The comprehensive numerical analysis reveals that the spatial distribution and directional components of the magnetic field within the Halbach array are highly sensitive to both geometric configuration and spatial distance. The field characteristics exhibit strong spatial confinement near the array surface, dominated by the B_x_ and B_z_ components, while the B_y_ component remains negligible, confirming the two-dimensional nature of the magnetic modulation within the X–Z plane. The monotonic and approximately linear variation of B_x_ with rotation angle and axial distance provides a robust physical basis for accurate distance sensing and angular calibration, particularly in systems employing decoupled soft magnetic films and rigid Hall sensors.

The analysis of the Z-directional field behavior further demonstrates a progressive transition from localized, nonlinear distributions near the surface to quasi-uniform fields in the far region, accompanied by an exponential attenuation in field magnitude. This transition implies a natural trade-off between sensing sensitivity and range, which can be tuned through the array geometry. The comparative study of different array diameters confirms that while the overall field intensity scales inversely with array size, the modulation characteristics, such as linearity, monotonicity, and confinement, remain invariant, reflecting the inherent scalability and universality of the Halbach configuration.

Importantly, these results indicate that optimizing the array diameter can significantly enhance spatial resolution and SNR without compromising the field’s structural integrity. The findings also suggest that deep learning-based models could exploit the nonlinear spatial correlations observed in B_x_ and B_z_ to achieve higher accuracy in multi-axis force or deformation sensing, paving the way for adaptive calibration strategies in soft magnetic sensor systems. To clearly situate our work, Table 2 provides a comparative analysis against other state-of-the-art magnetic sensing methods.

In addition, experimental validation represents the next critical step. We are currently fabricating and testing physical prototypes to confirm the predicted field structures and evaluate sensing performance. The practical implementation will require precise magnet alignment and mechanical stabilization during fabrication. Furthermore, as real deformation induces complex six-degree-of-freedom motion, we plan to adopt a data-driven calibration strategy combining multi-axis force measurements with machine learning to map the nonlinear relationship between magnetic signals and external stimuli. Leveraging the high SNR and structured field distribution of the Halbach design, these efforts aim to enable robust, wide-range, and high-precision tactile sensing for soft robotics and intelligent systems.

While this numerical study establishes the theoretical foundation for a Halbach-array-based magnetic tactile sensor, several limitations remain. The present model isolates in-plane magnet rotation to clarify its role in field modulation, but it does not yet account for coupled deformation modes, such as translational and out-of-plane displacements, or the nonlinear mechanical behavior of the elastomer substrate. These factors will inevitably influence the real sensor’s magnetic response and will be incorporated in future coupled magneto-mechanical simulations.

## 5. Conclusions

This study presents a comprehensive numerical investigation of a Halbach-array-based magnetic tactile sensor, establishing a theoretical framework for a design that enables directional magnetic field modulation and remote, decoupled sensing. The simulations demonstrate that the k = 2 Halbach configuration, combined with N52 permanent magnets, generates a high-strength magnetic field exceeding 1 mT at a remote sensing distance of 15 mm. Crucially, the array successfully tailors the magnetic field, reducing its spatial dimensionality from a complex 3D profile to a quasi-1D distribution, where the transverse components (B_y_ and B_z_) are suppressed by more than two orders of magnitude relative to the dominant B_x_ component near the central axis.

Furthermore, the sensor demonstrates a highly stable and predictable response to magnet rotation. The field distribution remains spatially consistent, with the rotation primarily modulating the field’s magnitude rather than its direction. This response, measured by the peak B_x_ amplitude and position, exhibits excellent linearity (R^2^ > 0.99) as a function of rotation angle. Finally, the parameter study establishes a clear design guideline, indicating that an array diameter of 10–20 mm provides the optimal balance by maximizing spatial resolution while avoiding the signal ambiguity observed in larger arrays. Collectively, these quantitative findings confirm that the Halbach-array-based design offers a robust physical foundation for high-precision, wide-range, and remote tactile sensing applications.

## Figures and Tables

**Figure 1 sensors-25-07240-f001:**
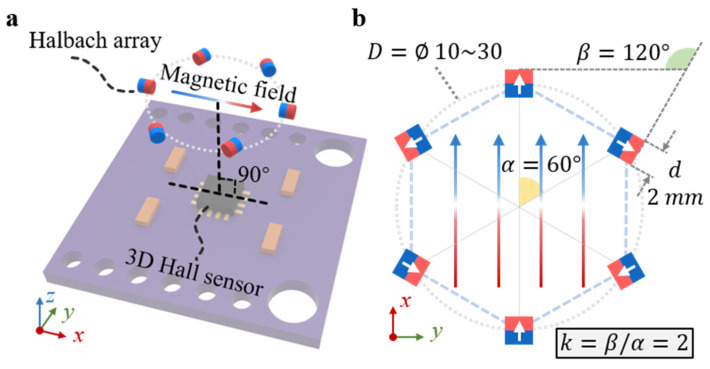
Schematic illustration of the proposed Halbach-array-based magnetic tactile sensor. (**a**) The sensor comprises a magnetic film formed by embedding patterned rigid small magnets within a soft elastomer matrix. A three-dimensional Hall sensor is positioned beneath the array to continuously measure the magnetic field variations, which are subsequently used to estimate both the direction and magnitude of applied mechanical forces on the elastomer surface. (**b**) Top view of the Halbach array consisting of 6 permanent magnets arranged in a k = 2 configuration. The modulated magnetic field within the array is predominantly oriented along the x-axis, while the field components along the y- and z-axes are significantly suppressed, resulting in a directional magnetic field distribution.

**Figure 2 sensors-25-07240-f002:**
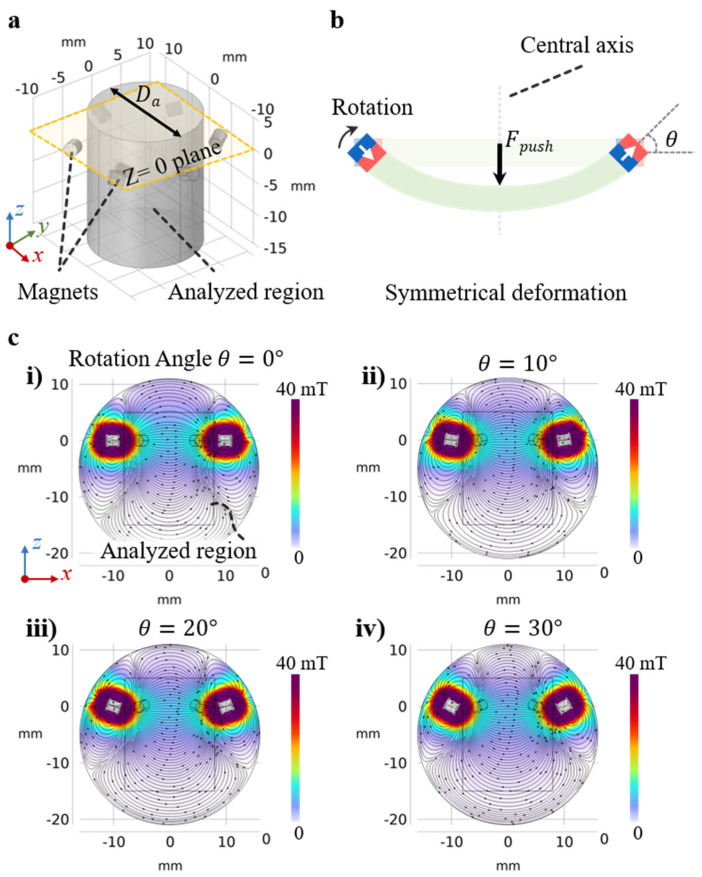
Finite element modeling of the proposed Halbach array for large-deformation tactile sensing. (**a**) The Halbach array is composed of six cylindrical magnets, each with a diameter of 2 mm and a height of 2 mm. The simulations were performed using the “Magnetic Fields, No Currents” module in COMSOL Multiphysics, incorporating sintered NdFeB magnets and surrounding air domains. (**b**) Schematic of the assumed deformation mode. The applied force acts at the center of the array, and the resulting magnet motion is simplified as in-plane rotation about their central axes, without translational displacement. (**c**) Simulated magnetic field distribution in the x–z plane (where y = 0 mm), showing the magnetic flux orientation and field modulation under varying rotation angles. The color scale is clamped to a maximum value of 40 mT to better visualize the magnetic field distribution within the analyzed region.

**Figure 3 sensors-25-07240-f003:**
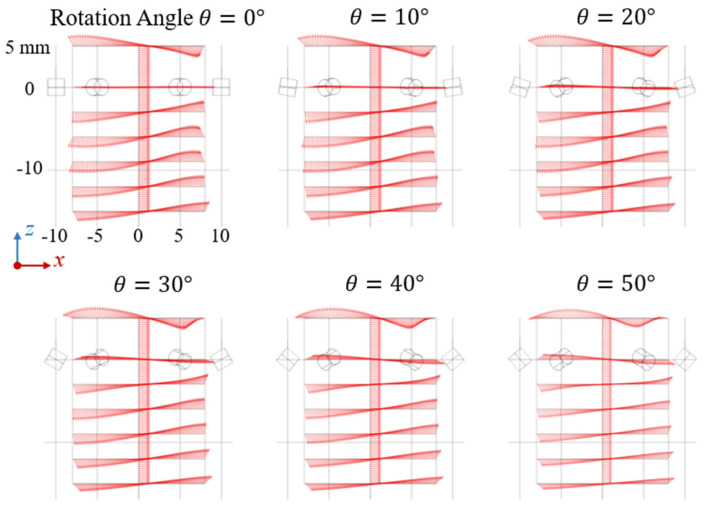
Directional profile of the magnetic field generated by the Halbach array in the x–z plane. Along the central vertical line, the magnetic field is oriented entirely along the x-axis. The field directions along the horizontal lines exhibit a consistent distribution pattern as the array rotates from 0° to 50°. The arrow lines are uniform in length to only demonstrate the directional feature.

**Figure 4 sensors-25-07240-f004:**
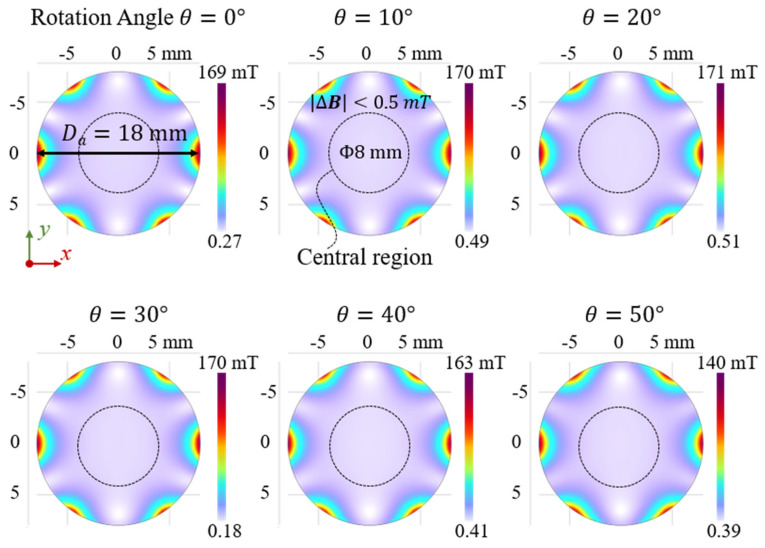
Magnetic flux density distribution within the Halbach array at z = 0 mm. As the magnets rotate from 0° to 50°, the overall field pattern remains stable, showing no significant change in spatial distribution apart from a slight decrease in absolute flux density.

**Figure 5 sensors-25-07240-f005:**
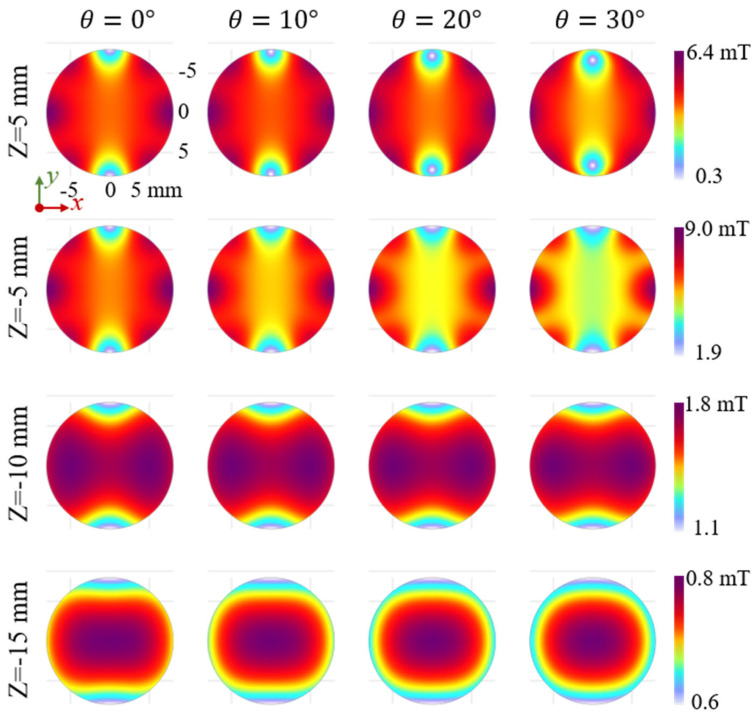
Simulated magnetic flux density distribution in horizontal cross-sections of the Halbach array at varied probe heights (z = 5, −5, −10, −15 mm) and rotation angles (*θ* = 0°–30°). The results show that smaller |z| yields higher flux density and steeper gradients, while rotation (*θ*) synchronously modulates the field orientation and distribution pattern, demonstrating the sensing ability of applied force.

**Figure 6 sensors-25-07240-f006:**
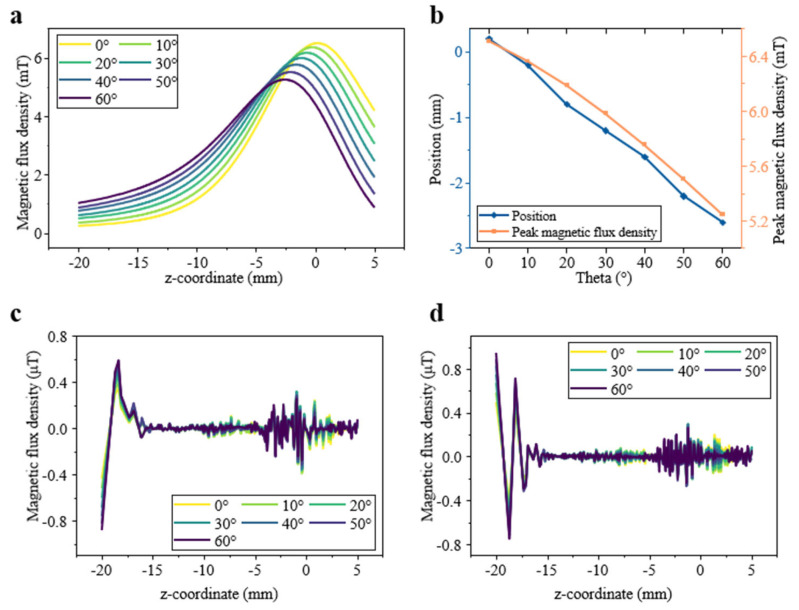
Magnetic flux density distribution at central Z-axis and the angular dependence. (**a**) magnetic flux density (in mT) along the z-coordinate (mm) for *θ* values ranging from 0° to 60°. (**b**) the position and strength change in peak magnetic flux density as *θ* increase from 0° to 60°. (**c**,**d**) magnetic flux density (in μT) along the z-coordinate.

**Figure 7 sensors-25-07240-f007:**
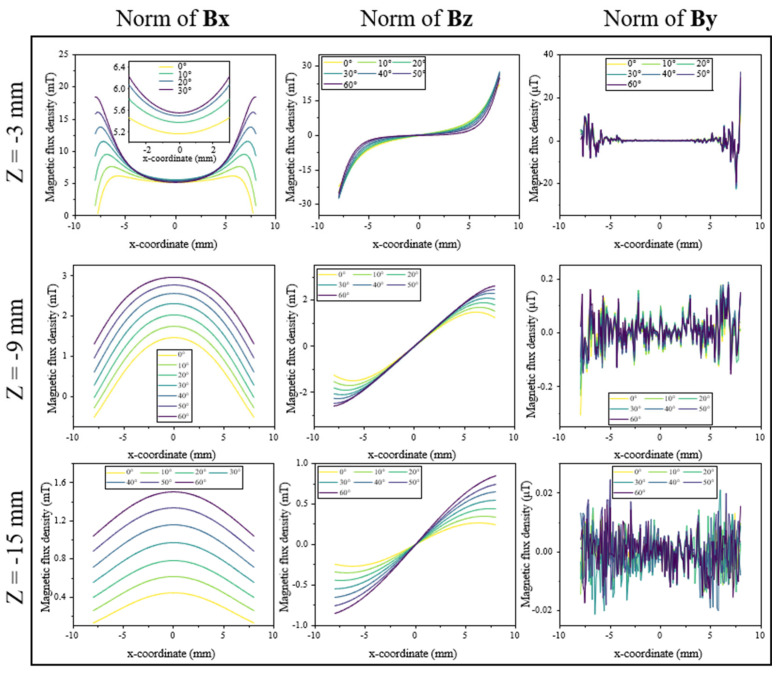
Magnetic flux density and its directional components along the X-axis at various Z-axis heights within the Halbach array. Each column corresponds to a field component (B_x_, B_y_, and B_z_), and each row represents a different Z position. The curves depict magnetic flux density distributions along the X-axis from −8 mm to 8 mm at multiple rotation angles. The inset images provide an enlarged view of the central region (−3 mm to 3 mm).

**Figure 8 sensors-25-07240-f008:**
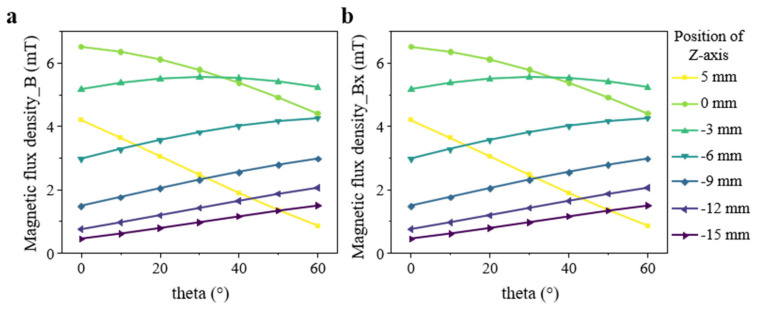
Quantitative analysis of magnetic flux density at the center of the Halbach array (X = 0 mm, Y = 0 mm). (**a**,**b**) the total magnetic flux density (B) and its X-directional component (B_x_) are plotted as functions of rotation angle (*θ*) and axial position (Z), while *θ* ranges from 0° to 60°, and Z-coordinate range from 5 mm to −15 mm.

**Figure 9 sensors-25-07240-f009:**
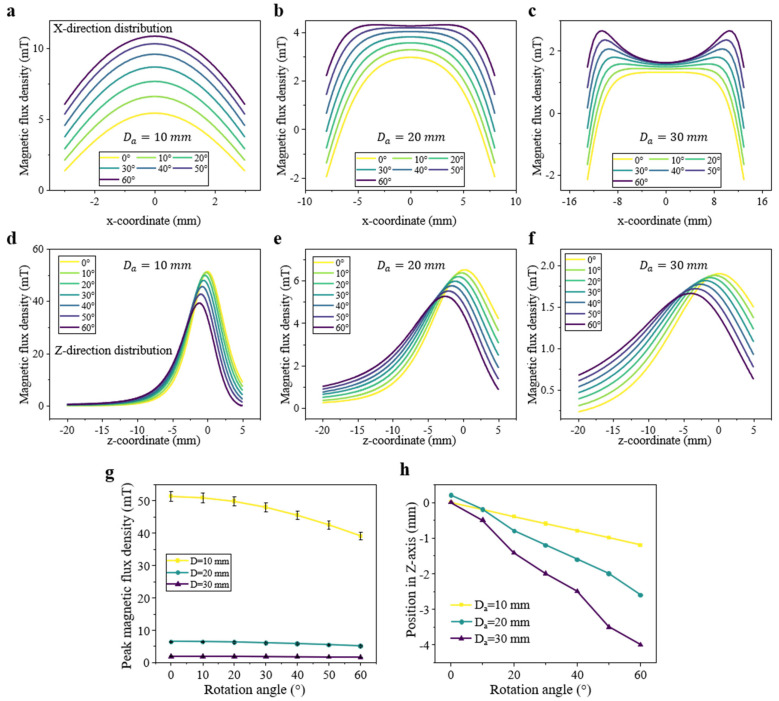
Simulated magnetic field characteristics of Halbach arrays with different diameters (10 mm, 20 mm, 30 mm). (**a**–**c**) Simulated magnetic field distributions of the B_x_ component in the x–y plane at Z = −6 mm, y = 0 mm. (**d**–**f**) Simulated magnetic field distributions along the Z direction, ranging from 5 mm to −20 mm. (**g**,**h**) Quantitative analysis from the simulation data, showing the Z-directional magnetic field peak intensity and position.

**Table 1 sensors-25-07240-t001:** Average magnetic flux densities and signal dominance ratios.

Distance	Avg. B_x_ (mT)	Avg. B_y_ (mT)	Order of (B_x_/B_y_)
−3 mm	5.81±0.43	$-6.37\times 10^{-7}\pm 4.24\times 10^{-6}$	$\sim 10^{7}$
−9 mm	1.48±0.25	$5.36\times 10^{-5}\pm 3.37\times 10^{-4}$	$\sim 10^{4}$
−15 mm	0.55±0.05	$2.54\times 10^{-6}\pm 4.09\times 10^{-5}$	$\sim 10^{5}$

X-coordinate is ranging from −5 to 5 mm.

**Table 2 sensors-25-07240-t002:** Comparative analysis of representative magnetic tactile sensing methods.

Method	Magnetic Source	Field Configuration	Sensing Principle	Signal Strength (at >10 mm)
Hellebrekers (2019) [42]	Soft composite (particles)	Random particle distribution	Single 3-axis Magnetometer	μT-level (inferred)
Yan (2021) [47]	Soft composite (film)	Sinusoidal magnetization	3-axis Hall Sensor	μT-level
Becker (2022) [51]	(External magnet)	3D micro-origami cube	Anisotropic Magnetoresistance (AMR)	(N/A)
Rehan (2022) [36]	Discrete permanent magnets	4-magnet simple array	4x single-axis Hall Sensors	≈7.3 mT at 6 mm
Dai (2024) [35]	Soft composite (film)	Centripetal magnetization	3-axis Hall Sensor	≈40 μT at 20 mm
Our Proposed Work	Discrete permanent magnets (N52)	Halbach Array	3-axis Hall Sensor	≈1 mT at 15 mm

## Data Availability

The original contributions presented in this study are included in the article. Further inquiries can be directed to the corresponding authors.

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
