# Peer review of "Numerical Investigation of Halbach-Array-Based Flexible Magnetic Sensors for Wide-Range Deformation Detection"

_sensors, 2025, doi:10.3390/s25237240_

Round 1

Reviewer 1 Report

Comments and Suggestions for Authors

Summary:

The paper defines a method of simulating a Halbach-array-based magnetic tactile sensor, validating the dimensional reduction of the magnetic flux distribution. The main contribution is the enhanced magnetic field strength with the reduced special dimensionality. COMSOL was used to simulate the Halbach configuration with different variations in z-distance and magnet-rotation angle to show the stable nature of this design.

Comments:

1. The authors have a nice, systematic approach to look at how the mechanical and electrical domains of this sensing system are decoupled. However, while this paper shows a more detailed understanding of the decoupled architecture, there should be more in the introduction explaining previous work with decoupled mechanical (Dai et al. 2023 Advanced Materials, Sundaram et al. 2022 IEEE TRO) with similar magnetic profiles (Yan et al. 2021 Science Robotics). These bodies of work also use soft magnetic materials, which, while they are not as strong as the permanent magnets, some of these systems were still able to sense changes in deformation at distances greater than 15mm with decent resolution.

  1. The authors mention that this paper establishes the theoretical framework and future work will show the experimental test results, but this current paper would benefit from experimental validation. The COMSOL simulations seem theoretically logical, but supporting these results with experimental testing would significantly strengthen this work. While the simulations can isolate a dominant mode of deformation-induced changes in magnetic flux distribution, the experimental data will also capture the nonlinearities of the soft material and the complexities of the deformation/ magnet movements, and this would be interested to compare with the simulation data.
  2. Figure comments:
  • Fig 1 – the magnets are depicted as cubes, but modeled using a cylindrical shape, so the figure should mirror this
  • Fig 2a – the magnets are positioned at the z=0 plane and should be shown in a clearer way.
  • Fig 2c – the color scale should be modified… currently, the permanent magnetic flux distribution is being emphasized, rather than the analyzed sensing region (which I assume is what the focus should be). The visuals should focus on the field in the measurement zone.
  • Fig 5 – the simulated magnetic flux density distribution at z = 5mm is larger than expected, is there a reason for this?
  1. The authors tend to repeat the stability of Halbach array magnetic field configuration and By/Bz signal minimization statements. Consolidating the tests that lead to these conclusions and then discussing these points would improve readability.
  2. The practical relevance of focusing on the magnet rotation around its own axis should be highlighted, it’s currently left to the reader to understand on their own terms.
  3. “Axial heights” should be more specific… probe height?
  4. It would be valuable to include some justification for the base design that the authors present. Why those magnets? Why D = 10 – 30 dia? Why the 15 mm displacement range?

Author Response

Response to Reviews

We thank the reviewers for the helpful review. We have addressed all the comments in the revised manuscript and the response letter. The original reviews are included in this response letter, where the reviewers’ comments are in blue. Our responses are in black. The revised parts of the manuscript are included and highlighted in yellow. References are listed at the end of the response letter.

Reviewer #1:

Comments:

The paper defines a method of simulating a Halbach-array-based magnetic tactile sensor, validating the dimensional reduction of the magnetic flux distribution. The main contribution is the enhanced magnetic field strength with the reduced special dimensionality. COMSOL was used to simulate the Halbach configuration with different variations in z-distance and magnet-rotation angle to show the stable nature of this design.

Response: We thank the reviewer for this thorough and constructive feedback on our manuscript. The comments are insightful and have helped us significantly improve the paper's clarity, context, and focus. We have now revised the manuscript to address all points, which were detailed below.

1) The authors have a nice, systematic approach to look at how the mechanical and electrical domains of this sensing system are decoupled. However, while this paper shows a more detailed understanding of the decoupled architecture, there should be more in the introduction explaining previous work with decoupled mechanical (Dai et al. 2023 Advanced Materials, Sundaram et al. 2022 IEEE TRO) with similar magnetic profiles (Yan et al. 2021 Science Robotics). These bodies of work also use soft magnetic materials, which, while they are not as strong as the permanent magnets, some of these systems were still able to sense changes in deformation at distances greater than 15mm with decent resolution.

Response: We thank the reviewer for this insightful comment. We agree that the organization of our original introduction section did not sufficiently contextualize our work within these important developments in decoupled sensing and magnetic field profiling. We have now revised the Introduction to include and discuss these state-of-art works with more details, which will provide a clearer understanding of the research background of our work.

The works by Dai et al. and Sundaram et al. are excellent examples of mechanically decoupled systems, where the deformable magnetic layer is physically separated from the rigid sensing electronics. This “split-type” design is a key strategy for improving robustness, and as the reviewer correctly notes, Dai et al. demonstrated an impressive sensing distance of over 20 mm. Similarly, the work by Yan et al. is a seminal paper demonstrating how a profiled magnetic field can be used to achieve force self-decoupling.

Our work builds directly on these concepts but explores a distinct and complementary approach. The aforementioned studies typically rely on soft magnetic composites (i.e., magnetic powders dispersed in an elastomer). While highly flexible, these materials produce a weaker magnetic field, and the complex 3D field distributions (even in profiled form) often require complex models for 3D force decoupling or machine learning for super-resolution.

To make this positioning clear, we have revised the Introduction section to explicitly discuss these works. We have also added the Sundaram et al. paper as a new reference.

(Main manuscript) “First, as noted in designs by Yan et al. [42], when the Hall sensor is directly attached behind the magnetic film, the rigid-soft interface restricts the mechanical de-formability, leading to nonlinear response under large strain. To address this, ‘split-type’ or decoupled architectures have been proposed, such as by Dai et al. [32] and Sundaram et al. [43], which physically separate the deformable magnetic layer from the rigid sensor. These designs improve robustness and enable remote sensing. Second, the magnetic field from conventional magnets is non-uniform and three-dimensional, complicating calibration. While soft magnetic composites allow for novel field profiling (e.g., sinusoidal or centripetal) to achieve force decoupling, the resulting 3D fields often require complex analytical models or machine learning for interpretation.”

2) The authors mention that this paper establishes the theoretical framework and future work will show the experimental test results, but this current paper would benefit from experimental validation. The COMSOL simulations seem theoretically logical, but supporting these results with experimental testing would significantly strengthen this work. While the simulations can isolate a dominant mode of deformation-induced changes in magnetic flux distribution, the experimental data will also capture the nonlinearities of the soft material and the complexities of the deformation/ magnet movements, and this would be interested to compare with the simulation data.

Response: We thank the reviewer for this important and constructive comment. We completely agree that experimental validation is the gold standard and would significantly strengthen this work. The reviewer makes an excellent point that an experimental system would capture complex, real-world behaviors, such as the material nonlinearities of the PDMS and coupled deformation modes (e.g., translation and rotation), which our current, simplified model does not.

In this context, we have intentionally scoped this specific manuscript as a foundational numerical investigation. The primary contribution is to establish a theoretical framework. A key advantage of a simulation-based approach is that it allows for a systematic and continuous exploration of the entire spatial distribution of the magnetic field, which is impractical to achieve with high fidelity in a physical experiment. This numerical analysis, therefore, serves as a critical design guide for subsequent experimental work, particularly for optimizing Hall sensor placement and informing signal processing strategies. We believe this theoretical exploration is a necessary prerequisite to build a robust sensor and holds value for other researchers in the field.

We also wish to explain that this is not just a theoretical exercise. Our team is actively pursuing the experimental validation of this concept. We have already fabricated the flexible Halbach array samples and are in the process of integrating them with our 3D Hall sensing unit. Our ongoing work involves collecting experimental data and training a deep learning model to leverage the unique field characteristics identified in this paper.

To make this scope and the path forward clearer to the audiences, we have substantially revised the Conclusions section to more explicitly state the limitations of the current numerical-only model and to detail our ongoing experimental work.

(Main manuscript) “While this numerical study establishes the theoretical foundation for a Hal-bach-array-based magnetic tactile sensor, several limitations remain. The present model isolates in-plane magnet rotation to clarify its role in field modulation, but it does not yet account for coupled deformation modes, such as translational and out-of-plane dis-placements, or the nonlinear mechanical behavior of the elastomer substrate. These factors will inevitably influence the real sensor’s magnetic response and will be incorporated in future coupled magneto-mechanical simulations.”

3) Figure comments:

Fig.1 – the magnets are depicted as cubes, but modeled using a cylindrical shape, so the figure should mirror this.

Response: We thank the reviewer for pointing out the inconsistency. Our intention was to use cubes in the schematic for simpler visual illustration, while the simulation model correctly used cylindrical magnets as this is a common, commercially available shape. We agree that this discrepancy is confusing. To ensure consistency throughout the manuscript, we have revised Figure 1. The schematic now depicts cylindrical magnets instead of cubes, which perfectly aligns with the model parameters detailed in Section 2.2.

Figure 1. Schematic illustration of the proposed Halbach-array-based magnetic tactile sensor.

Fig.2a – the magnets are positioned at the Z=0 plane and should be shown in a clearer way.

Response: We thank the reviewer for this suggestion. We agree that the visualization of the simulation geometry in the original Figure 2a was not sufficiently clear.

We have now revised Figure 2a to better illustrate the model. The Z=0 mm plane is now rendered with highlight and necessary description is added, making the position of the six cylindrical magnets and the central analysis zone clearly visible.

Fig. 2c – the color scale should be modified… currently, the permanent magnetic flux distribution is being emphasized, rather than the analyzed sensing region (which I assume is what the focus should be). The visuals should focus on the field in the measurement zone.

Response: We sincerely thank the reviewer for this suggestion. The reviewer is correct that the high magnetic flux density at the surface of the magnets themselves (approaching 1.44 T) saturates the color scale, which completely obscures the more subtle, mT-level field variations within the “analyzed region” below the array. This is, as the reviewer notes, the actual region of interest for sensing.

To correct this, we have regenerated Figure 2c with a clamped color scale. The color bar maximum has been set to 40 mT that is appropriate for the sensing region, which now clearly visualizes the shape, uniformity, and magnitude of the magnetic field in the area where a Hall sensor would be placed. We have also added a note to the figure captions to explain this adjustment.

Figure 2. Finite element modeling of the proposed Halbach array for large-deformation tactile sensing. (A) The Halbach array is composed of six cylindrical magnets, each with a diameter of 2 mm and a height of 2 mm. The simulations were performed using the “Magnetic Fields, No Currents” module in COMSOL Multiphysics, incorporating sintered NdFeB magnets and surrounding air domains. (B) Schematic of the assumed deformation mode. The applied force acts at the center of the array, and the resulting magnet motion is simplified as in-plane rotation about their central axes, without translational displacement. (C) Simulated magnetic field distribution in the x–z plane (where y = 0 mm), showing the magnetic flux orientation and field modulation under varying rotation angles. The color scale is clamped to a maximum value of 40 mT to better visualize the magnetic field distribution within the analyzed region.

Fig 5 – the simulated magnetic flux density distribution at Z=5mm is larger than expected, is there a reason for this?

Response: We thank the reviewer for this insightful question, as it allows us to clarify an important aspect of the Halbach array's field distribution. We would like to clarify that the field at z = +5 mm is, in fact, weaker than the field at the corresponding plane in the intended sensing region (z = -5 mm). As shown in Figure 5 (top row), the maximum flux density at z = +5 mm is approximately 6.4 mT, whereas at z = -5 mm (second row), the maximum is approximately 9.0 mT.

This asymmetric distribution is a fundamental property of the k=2 Halbach array while magnets rotation, which is designed to enhance the magnetic field on one side (our intended sensing side, z < 0) and suppress it on the opposite side (z > 0), corresponding to the direction of applied force. The field at z = +5 mm is this “suppressed” field, which is non-zero but weaker than the “enhanced” field at z = -5 mm. The reviewer's observation that this field still appears strong is astute. While our primary design places the Hall sensor in the enhanced z < 0 region, the fact that a measurable, well-structured field exists at z = +5 mm (as also seen in the flux lines of Figure 2C) implies that a sensor could, in principle, be placed above the magnetic film to detect deformation, opening possibilities for alternative sensor configurations.

We recognize that this was not clearly explained in the original text. We have revised Section 3.2 to explicitly differentiate between the enhanced sensing region (z < 0) and the suppressed region (z > 0) and to clarify this asymmetric behavior.

(Main manuscript) “It is critical to note that the sensor is designed to be placed below the array (in the negative z-direction). The top row (z = 5 mm) shows the magnetic field on the suppressed (weak) side of the Halbach array. The subsequent rows (z = -5 mm to -20 mm) show the field in the enhanced (strong) sensing region.”

4) The authors tend to repeat the stability of Halbach array magnetic field configuration and By/Bz signal minimization statements. Consolidating the tests that lead to these conclusions and then discussing these points would improve readability.

Response: We thank the reviewer for this helpful comment. We have now removed several repeatable statements on the stability of Halbach array configuration and By/Bz signal minimization, and summarized the findings in the conclusion section with proper discussions.

(Main manuscript) “The results presented in Figures 4-6 collectively demonstrate the intrinsic stability and directional selectivity of the Halbach array’s magnetic field configuration. Across multiple cross-sectional analyses, the field distribution remains consistent under varying rotational angles, confirming that magnet rotation primarily modulates field magnitude rather than direction. Moreover, the transverse components ?y and Bz remain several orders of magnitude lower than Bx, particularly near the central axis, indicating effective suppression of cross-axis coupling. This intrinsic one-dimensional field orientation simplifies sensor calibration, enhances signal-to-noise ratio, and ensures reliable correlation between magnetic field variation and mechanical input. These findings establish the Halbach array as a robust magnetic source with stable spatial characteristics, forming the physical foundation for high-resolution and directionally decoupled tactile sensing.”

5) The practical relevance of focusing on the magnet rotation around its own axis should be highlighted, it’s currently left to the reader to understand on their own terms.

Response: We thank the reviewer for this valuable suggestion. In the revised manuscript, we have clarified the practical significance of modeling the magnet rotation around its own axis. In the proposed tactile sensing configuration, external pressure or shear applied to the soft magnetic film primarily induces local bending and torsion, which result in small-angle rotations of the embedded magnetic elements. The translational motion of the magnets, on the other hand, is relatively limited due to the mechanical constraint of the elastomeric matrix. Therefore, rotation around the central axis represents the dominant deformation mode governing the modulation of the magnetic field detected by the sensor array.

This modeling focus allows us to isolate and analyze the primary mechanism responsible for magnetic signal variation while maintaining computational clarity. We have added a corresponding explanation in the revised manuscript (section 2.2) to highlight the physical and engineering relevance of this simplification. Other deformation modes, including vertical displacement and off-axis motion, will be incorporated in future work to achieve a more comprehensive description of the tactile response.

(Main manuscript) “To focus on the magnetic field variation induced solely by magnet rotation, several simplifications were introduced in the model. As shown in Figure 2B, all magnets were assumed to remain in the same plane and to rotate about their central axes without translational motion. The rotational angle, denoted as θ, was varied up to a maximum of 60°, enabling direct analysis of magnetic field modulation within the Halbach array as a function of magnet rotation.

This focus on magnet rotation around its own axis is not merely a theoretical simplification but corresponds closely to the dominant deformation mode in the proposed tactile sensing structure. In practical operation, external mechanical stimuli applied to the soft magnetic film mainly induce localized bending or shearing, which results in small-angle rotations of the embedded magnetic elements, while translational displacements remain relatively constrained by the elastomeric matrix. This rotation-driven modulation of the magnetic field thus represents the primary mechanism governing signal variation in the magnetic sensor array, directly influencing its sensitivity and linearity.

Nonetheless, these simplifications inevitably neglect out-of-plane displacements and relative positional shifts of individual magnets that can arise when the magnetic elastomer film deforms under external loading. Such effects are expected to influence the local magnetic flux distribution and may introduce nonlinear coupling between mechanical deformation and magnetic response. Therefore, while the present model effectively isolates the fundamental role of magnet rotation in magnetic field modulation, it does not yet fully capture the coupled magneto-mechanical behavior of the system. These additional deformation modes, including vertical displacement and magnet–matrix detachment, will be quantitatively incorporated in future simulations and experimental validations to achieve a more comprehensive understanding of the sensor’s response mechanism.”

6) “Axial heights” should be more specific… probe height?.

Response: We appreciate the reviewer’s insightful suggestion. We agree that the term “axial heights” may cause ambiguity and that “probe height” provides a clearer and more precise description of the measurement context. Accordingly, we have revised all relevant instances in the manuscript to use “probe height” instead of “axial height.” This correction improves clarity and better reflects the physical meaning of the measurement process in our analysis.

7) It would be valuable to include some justification for the base design that the authors present. Why those magnets? Why D = 10 – 30 dia? Why the 15 mm displacement range?

Response: We thank the reviewer for this helpful comment. We agree that the justification for these key design parameters was missing from the original manuscript and is important for contextualizing the study.

We have now added a new paragraph in Section 2.2 (“Finite Element Method for Sensor Modeling”) to provide this rationale, which is based on balancing application requirements with practical design trade-offs.

Magnet Choice (N52, 2mm x 2mm): We selected sintered NdFeB (N52) magnets as they provide the highest remanent flux density (approx. 1.44 T) in a compact, commercially available form factor. The 2mm size is suitable for embedding within a soft PDMS substrate while being large enough to be practically assembled into an array.

Array Diameter (D = 10 - 30 mm): This range was chosen to match the typical scale of a single ‘taxel’ (tactile pixel) in robotic fingertips or e-skin applications. As our analysis in Section 3.4 shows, there is a trade-off between spatial resolution (which is enhanced at smaller diameters) and the practical challenges of assembling smaller, more complex arrays. The 10-30 mm range represents a practical and relevant design space.

Sensing Range (up to 15 mm): This distance was a key parameter of our investigation. A primary goal was to demonstrate that our proposed Halbach array, using strong permanent magnets, could achieve true 'remote' or 'decoupled' sensing. We aimed to verify that, at this significant 15 mm distance, the sensor could still detect a millitesla (mT)-level signal. This is a critical differentiator from many soft magnetic composite sensors, which often only provide microtesla (uT)-level signals at such distances, resulting in a much lower signal-to-noise ratio (SNR). Our simulations confirm the Halbach design achieves this, maintaining a strong signal for high-fidelity sensing.

We have incorporated these justifications into the revised manuscript.

(Main manuscript) “The baseline design parameters were chosen to be representative of tactile sensing applications, balancing performance with practical implementation. The individual magnets (sintered NdFeB N52, 2mm diameter, 2mm height) were selected as they provide the highest remanent flux density (approx. 1.44 T) in a compact, commercially-available form factor suitable for embedding in a soft substrate. The array diameter (D) range of 10-30 mm was chosen to match the typical scale of a single tactile pixel for robotic fingertips and e-skin. This range also represents a practical trade-off between spatial resolution, which our analysis in Section 3.4 shows flux density is enhanced at smaller diameters, and the physical challenges of precisely assembling the array. The remote sensing distance (analyzed up to z = -15 mm) was a key parameter of this investigation. A primary objective was to verify that the Halbach array could maintain a strong, millitesla (mT)-level signal at this significant distance. This contrasts with many soft-composite-based sensors, which often drop to the microtesla (uT) level at similar distances, resulting in a much lower signal-to-noise ratio (SNR).”

Reviewer 2 Report

Comments and Suggestions for Authors

The authors have proposed a Halbach-array-based flexible magnetic sensor for wide-range deformation detection through numerical investigation. The study presents an interesting approach with potential applications in soft robotic sensing. However, the following comments should be addressed before the paper can be considered for acceptance.

  1. The authors claim that their method enables wide-range deformation detection. However, the simulations only consider the rotation of the magnet, not axial deformation. The authors should explicitly mention this limitation or clarify the relationship between deformation and magnetic rotation.
  2. The authors state that the proposed sensing method exhibits high sensitivity. However, in Figure 4, there appears to be no significant difference in the magnetic flux distribution between 0° and 30°. Moreover, the maximum flux density remains around 170 mT for all four cases. The authors should either revise this claim or provide a clear explanation supporting their assertion of high sensitivity.
  3. In Figure 5, there is a noticeable difference in the magnetic field distribution, particularly between 30° and 40°. Additionally, a 2.6 mT difference in maximum flux density is observed. Theoretically, the flux density should remain similar, as both represent 5 mm offset from the original position. The authors are requested to explain the reason for this discrepancy.
  4. Previous research, such as the Split-Type Magnetic Soft Tactile Sensor with 3D Force Decoupling, has demonstrated structural decoupling by separating the sensing electronics from the soft magnetic layer to achieve three-axis force decoupling. The authors should clearly articulate the novelty of their proposed work in comparison to these existing approaches. A comparative analysis with other magnetic sensing methods would further strengthen the contribution.
  5. Since the present study is entirely simulation-based, the authors should explicitly discuss the potential challenges that need to be addressed for experimental validation. Additionally, they should elaborate on how the proposed sensing method could be practically implemented to estimate deformation and interaction forces, which would enhance its relevance for real-world applications.
  6. The authors are encouraged to provide a more detailed explanation of the design parameter selection (e.g., cylinder diameter, magnet sizes, spacing, and other geometric factors) to help readers understand the design rationale.
  7. The manuscript contains several typographical and grammatical errors that need to be corrected for clarity and professionalism.

Author Response

Response to Reviews

We thank the reviewers for the helpful review. We have addressed all the comments in the revised manuscript and the response letter. The original reviews are included in this response letter, where the reviewers’ comments are in blue. Our responses are in black. The revised parts of the manuscript are included and highlighted in yellow. References are listed at the end of the response letter.

Reviewer #2:

Comments:

The authors have proposed a Halbach-array-based flexible magnetic sensor for wide-range deformation detection through numerical investigation. The study presents an interesting approach with potential applications in soft robotic sensing. However, the following comments should be addressed before the paper can be considered for acceptance.

Response: We sincerely thank the reviewer for the constructive and insightful comments. We have carefully revised the manuscript in accordance with the suggestions provided. The detailed, point-by-point responses to each comment are presented below.

  1. The authors claim that their method enables wide-range deformation detection. However, the simulations only consider the rotation of the magnet, not axial deformation. The authors should explicitly mention this limitation or clarify the relationship between deformation and magnetic rotation.

Response: We thank the reviewer for this critical comment. We agree that the relationship between the "wide-range deformation" of the soft film and the "magnet rotation" simulated in our model was not sufficiently explained. Our current numerical model is a simplification to focus on the influence of magnets rotation. In a real-world scenario, a deformation applied to the soft PDMS film would result in a complex, coupled motion of the embedded magnets, including translation (x, y displacement), axial displacement (z displacement), and rotation.

The primary goal of this foundational study was to isolate and systematically analyze one of these key components, local magnet rotation. As depicted in Figure 2B, this in-plane rotation is intended to serve as a first-order proxy for the local tilting that occurs when the elastomer is subjected to shear, torsional, or complex indentation forces. By focusing only on rotation, we can establish a clear baseline understanding of how this specific degree of freedom modulates the Halbach-configured field, which is a necessary step before introducing more complex, coupled motion.

We have now explicitly acknowledged this simplification and its implications in the revised manuscript. We have added a clarification in Section 2.2 (where the model is introduced) and have also expanded the Conclusions (Section 5) to state this as a key limitation and a direction for future work.

(Main manuscript, Section 2.2) “To focus on the magnetic field variation induced solely by magnet rotation, several simplifications were introduced in the model. As shown in Figure 2B, all magnets were assumed to remain in the same plane and to rotate about their central axes without translational motion. The rotational angle, denoted as θ, was varied up to a maximum of 60°, enabling direct analysis of magnetic field modulation within the Halbach array as a function of magnet rotation.

This focus on magnet rotation around its own axis is not merely a theoretical simplification but corresponds closely to the dominant deformation mode in the proposed tactile sensing structure. In practical operation, external mechanical stimuli applied to the soft magnetic film mainly induce localized bending or shearing, which results in small-angle rotations of the embedded magnetic elements, while translational displacements remain relatively constrained by the elastomeric matrix. This rotation-driven modulation of the magnetic field thus represents the primary mechanism governing signal variation in the magnetic sensor array, directly influencing its sensitivity and linearity.

Nonetheless, these simplifications inevitably neglect out-of-plane displacements and relative positional shifts of individual magnets that can arise when the magnetic elastomer film deforms under external loading. Such effects are expected to influence the local magnetic flux distribution and may introduce nonlinear coupling between mechanical deformation and magnetic response. Therefore, while the present model effectively isolates the fundamental role of magnet rotation in magnetic field modulation, it does not yet fully capture the coupled magneto-mechanical behavior of the system. These additional deformation modes, including vertical displacement and magnet-matrix detachment, will be quantitatively incorporated in future simulations and experimental validations to achieve a more comprehensive understanding of the sensor’s response mechanism.”

(Main manuscript, Section 4) “While this numerical study establishes the theoretical foundation for a Halbach-array-based magnetic tactile sensor, several limitations remain. The present model isolates in-plane magnet rotation to clarify its role in field modulation, but it does not yet account for coupled deformation modes, such as translational and out-of-plane dis-placements, or the nonlinear mechanical behavior of the elastomer substrate. These fac tors will inevitably influence the real sensor’s magnetic response and will be incorporated in future coupled magneto-mechanical simulations.”

  1. The authors state that the proposed sensing method exhibits high sensitivity. However, in Figure 4, there appears to be no significant difference in the magnetic flux distribution between 0° and 30°. Moreover, the maximum flux density remains around 170 mT for all four cases. The authors should either revise this claim or provide a clear explanation supporting their assertion of high sensitivity.

Response: We thank the reviewer for pointing out this apparent contradiction. This is a critical point that we agree was not clearly explained in the original manuscript. The comment on Figure 4 is entirely correct: the field distribution in the plane of the magnets (z=0) shows high stability and minimal change with rotation.

We must clarify two key points:

  1. Purpose of Figure 4 vs. Figures 6 & 7: Figure 4 is intended to demonstrate the stability of the magnetic field source at z=0. This stability is a desirable property, not a lack of sensitivity. The actual sensing takes place in the remote region (z < 0), where the Hall sensor is located. The sensor’s response to rotation is demonstrated in Figure 6 and Figure 7. For example, Figure 6B shows a highly linear (R2=0.990) and predictable change in Bx peak amplitude as θ increases. This demonstrates a clear, measurable response to the rotational input at the remote sensing location.
  2. Definition of “High Sensitivity”: Our claim of high sensitivity refers to the high Signal-to-Noise Ratio (SNR). As shown in Figure 6A, our Halbach design with permanent magnets provides a strong millitesla (mT)-level signal even at a remote distance of 15 mm. This signal is 2-3 orders of magnitude stronger than the microtesla (μT)-level signals often reported for soft magnetic composite sensors at similar distances [1,2]. This exceptionally high-strength signal is critical for achieving high-fidelity, high-sensitivity measurements in a practical, noisy environment.

To prevent this confusion, we have revised the manuscript to more clearly differentiate the purpose of Figure 4 (source stability) from Figures 6 & 7 (remote sensing response) and to explicitly define “high sensitivity” in the context of high SNR.

(Main manuscript, Section 3.2) “In addition to the x–z plane, the field distribution in the plane of the magnets themselves (z = 0 mm) was examined, as shown in Figure 4. This analysis is important for understanding the stability of the magnetic field source, not the sensing response. The actual sensing region is located remotely in the z < 0 space”

(Main manuscript, Section 3.3) “A key claim of this work is that the Halbach configuration enables “high sensitivity” sensing. We must clarify this claim in the context of Signal-to-Noise Ratio (SNR). As quantified in our analysis (Figure 6A), the use of permanent magnets in a Halbach array generates a strong, millitesla (mT)-level signal, even at a remote 15 mm distance. This signal is 2-3 orders of magnitude stronger than the microtesla (μT)-level signals typically reported by sensors using soft magnetic composites at similar distances [32,45]. This high-strength signal is a critical advantage for practical applications, as it provides a high SNR, enabling robust and high-fidelity measurements even in noisy environments. The sensor's response is then clearly and linearly resolved, as shown in Figure 6B”.

  1. In Figure 5, there is a noticeable difference in the magnetic field distribution, particularly between 30° and 40°. Additionally, a 2.6 mT difference in maximum flux density is observed. Theoretically, the flux density should remain similar, as both represent 5 mm offset from the original position. The authors are requested to explain the reason for this discrepancy.

Response: We sincerely thank the reviewer for this helpful comment. This question highlights the central mechanism of our proposed sensing system, which we failed to explain clearly. The reviewer's analysis is correct, but the discrepancy is not a simulation error; it is the intended physical effect of the design.

We would like to clarify several key points based on the reviewer’s comment:

  1. Symmetry at Rest (θ = 0°): The reviewer’s theoretical expectation is perfectly correct for the initial state. As shown in the first column of Figure 5 (at θ = 0°), the magnetic field is symmetric about the z=0 plane. The flux density distributions at z = +5 mm and z = -5 mm are identical.
  2. Rotation-Induced Asymmetry: The “discrepancy” (e.g., the 2.6 mT difference) is a result of the magnet rotation. As the magnets rotate (e.g., from 0° to 40°), this initial symmetry is broken. The Halbach configuration is designed such that this rotation dynamically restructures and “steers” the magnetic field, concentrating the flux downward into the z < 0 sensing region while simultaneously suppressing the field in the z > 0 region. This shows the rotation (caused by deformation) actively modulates the field strength asymmetrically.
  3. Non-Linear Field Response: The reviewer also notes that the change in the distribution’s shape is particularly noticeable at larger angles (e.g., between 30° and 40°). This illustrates that the spatial field’s response to rotation is non-linear. This complex, non-linear behavior across the entire plane (which differs from the linear response at the central axis shown in Fig. 6B) is a key insight that would be very difficult to map experimentally.

To address all these concerns, we have revised Section 3.2 to properly explain this critical, dynamic, and asymmetric behavior.

(Main manuscript, Section 3.2) “A key finding of this analysis is the rotation-induced asymmetry of the magnetic field. At rest (θ = 0°), the field is symmetric about the z = 0 plane, as shown in the first column of Figure 5, where the distributions at z = +5 mm and z = -5 mm are identical. Once the magnets rotate, this symmetry is broken: the Halbach array actively reshapes the field, enhancing the flux in the sensing region (z < 0) while suppressing it above the array (z > 0). Consequently, the field at z = -5 mm becomes stronger than at z = +5 mm. As rotation increases (e.g., 30°-40°), the field distribution evolves nonlinearly, even though the axial peak flux varies linearly, revealing spatial complexity that underpins the sensor’s modulation behavior.”

  1. Previous research, such as the Split-Type Magnetic Soft Tactile Sensor with 3D Force Decoupling, has demonstrated structural decoupling by separating the sensing electronics from the soft magnetic layer to achieve three-axis force decoupling. The authors should clearly articulate the novelty of their proposed work in comparison to these existing approaches. A comparative analysis with other magnetic sensing methods would further strengthen the contribution.

Response: We thank the reviewer for this excellent comment, which helps us to precisely position the unique contribution of our work.

The reviewer pointed out that prior art, such as the excellent work by Dai et al., has successfully demonstrated the “split-type” or structurally decoupled approach. The novelty of our paper lies not in the concept of structural decoupling itself, but rather in the method of magnetic field generation and structuring within that decoupled framework.

The “Split-Type” sensor and other similar works typically employ soft magnetic composites (i.e., magnetic particles in an elastomer). While this allows for flexible, custom-profiled films, the resulting magnetic field strength at a remote sensing distance is often in the microtesla (μT) range. For example, the sensor signals are around approx. 40 uT at a 20 mm distance.

Our work investigates a fundamentally different approach: the use of discrete N52 permanent magnets arranged in a Halbach array. This design offers two distinct advantages:

(1) Massive Signal Strength: As shown in our analysis (e.g., Fig. 6A), this configuration generates a significantly stronger field, providing a millitesla (mT)-level signal (> 1 mT) even at a 15 mm distance. This is approximately 100- to 1000-fold stronger than the uT-level signals from composite-based sensors at similar distances. This provides a vastly superior Signal-to-Noise Ratio (SNR), which is a critical advantage for sensing accuracy, reducing signal processing overhead, and enabling sensing at even greater distances.

(2) Field Dimensionality Reduction: The Halbach array fundamentally reduces the magnetic field's dimensionality to a quasi-1D distribution a priori. This offers a different pathway to simplified signal processing compared to the complex 3D analytical model required to decouple the 3D centripetal field.

To further strengthen this contribution and explicitly address the reviewer’s request for a comparative analysis, we have now added a new comparative table (Table 1) in the Discussion section to clearly summarize these differences.

(Main manuscript, Section 4) “Importantly, these results indicate that optimizing the array diameter can significantly enhance spatial resolution and SNR without compromising the field’s structural integrity. The findings also suggest that deep learning-based models could exploit the nonlinear spatial correlations observed in Bx and Bz to achieve higher accuracy in multi-axis force or deformation sensing, paving the way for adaptive calibration strategies in soft magnetic sensor systems. To clearly situate our work, Table 1 provides a comparative analysis against other state-of-the-art magnetic sensing methods.”

Table 1. Comparative analysis of representative magnetic tactile sensing methods.

Method

Magnetic Source

Field Configuration

Sensing Principle

Signal Strength (at >10mm)

Hellebrekers (2019) [37]

Soft composite (particles)

Random particle distribution

Single 3-axis Magnetometer

μT-level (inferred)

Yan (2021) [42]

Soft composite (film)

Sinusoidal magnetization

3-axis Hall Sensor

μT-level

Becker (2022) [46]

(External magnet)

3D micro-origami cube

Anisotropic Magnetoresistance (AMR)

(N/A)

Rehan (2022) [33]

Discrete permanent magnets

4-magnet simple array

4x single-axis Hall Sensors

≈7.3 mT @ 6mm

Dai (2024) [32]

Soft composite (film)

Centripetal magnetization

3-axis Hall Sensor

≈40μT @ 20mm

Our Proposed Work

Discrete permanent magnets (N52)

Halbach Array

3-axis Hall Sensor

≈1mT @ 15mm

  1. Since the present study is entirely simulation-based, the authors should explicitly discuss the potential challenges that need to be addressed for experimental validation. Additionally, they should elaborate on how the proposed sensing method could be practically implemented to estimate deformation and interaction forces, which would enhance its relevance for real-world applications.

Response: We thank the reviewer for this constructive and forward-looking comment. The discussion on the practical transition from simulation to a physical device is essential for this work. To address these important points, we have added a new paragraph in the Discussion (Section 4) that explicitly details both the anticipated experimental challenges and the practical implementation workflow.

Experimental Challenges:

  1. Fabrication: The practical difficulty of precisely assembling and holding the six strong, mutually-repelling N52 magnets in the correct Halbach orientation within the soft PDMS matrix during its curing process.
  2. Complex Deformation: Our model’s simplification of deformation to pure rotation. We now state that a real-world deformation will induce a complex, coupled 6-DOF motion (translation in x, y, z and rotation in all three axes).
  3. Material Non-Linearity: The inherent hyperelastic and hysteretic properties of the PDMS substrate, which are not captured in the current model and will affect the force-to-deformation mapping.

Moreover, we have also elaborated on how the sensor would be implemented:

  1. A practical sensor would rely on a data-driven calibration approach. This involves using a ground-truth system (e.g., a 6-axis force sensor) to apply known forces and simultaneously recording the 3-axis magnetic vector (Bx, By, Bz) from the Hall sensor.
  2. A machine learning model then will be trained to learn the complex, non-linear mapping to ( ).

In the revised manuscript, we emphasize that the key advantages of our Halbach design, the exceptionally high SNR from the mT-level signal and the uniquely structured quasi-1D field, would provide a rich, high-quality input for such a model.

(Main manuscript, Section 4-Discussion) “While this numerical study establishes the theoretical foundation for a Halbach-array-based magnetic tactile sensor, several limitations remain. The present model isolates in-plane magnet rotation to clarify its role in field modulation, but it does not yet account for coupled deformation modes, such as translational and out-of-plane displacements, or the nonlinear mechanical behavior of the elastomer substrate. These factors will inevitably influence the real sensor’s magnetic response and will be incorporated in future coupled magneto-mechanical simulations.”

(Main manuscript, Section 5-Conclusion) “This study presents a comprehensive numerical investigation of a Halbach-array-based magnetic tactile sensor, demonstrating a design that enables directional magnetic field modulation and remote, decoupled sensing. The primary findings of this simulation study are:

  1. High Signal Strength: The k=2 Halbach configuration, combined with N52 permanent magnets, generates a high-strength, millitesla (mT)-level magnetic field, enabling robust remote detection at distance up to 15 mm.
  2. Field Dimensionality Reduction: The array successfully tailors the magnetic field, reducing its spatial dimensionality from a complex 3D profile to a quasi-1D distribution, as quantified in Table 1. The transverse components (By and Bz) are suppressed by nearly two orders of magnitude compared to Bx near the central axis.
  3. Linear & Stable Response: The sensor’s response to magnet rotation, a proxy for film deformation, is highly predictable. As shown in Figures 4-7, the field distribution remains spatially stable, with the rotation primarily modulating the field’s magnitude, not its direction. This response, measured by the peak Bx amplitude and position, exhibits excellent linearity (R2 > 0.99).
  4. Design Guideline: The parameter study (Figure 9) established a clear design trade-off: an array diameter of 10-20 mm provides the optimal balance, maximizing spatial resolution while avoiding the signal ambiguity seen in larger arrays.

Taken together, these findings establish a robust theoretical framework for this sensor. The intrinsic 1D field orientation and high SNR simplify calibration and form a solid physical foundation for the future development of high-precision, wide-range flexible magnetic sensors for robotics and wearable applications.”

  1. The authors are encouraged to provide a more detailed explanation of the design parameter selection (e.g., cylinder diameter, magnet sizes, spacing, and other geometric factors) to help readers understand the design rationale.

Response: We thank the reviewer for this helpful comment. We agree that the justification for these key design parameters was missing from the original manuscript and is important for contextualizing the study. We have now added a new paragraph in Section 2.2 (“Finite Element Method for Sensor Modeling”) to provide this rationale, which is based on balancing application requirements with practical design trade-offs.

Magnet Choice (N52, 2mm x 2mm): We selected sintered NdFeB (N52) magnets as they provide the highest remanent flux density (approx. 1.44 T) in a compact, commercially available form factor. The 2mm size is suitable for embedding within a soft PDMS substrate while being large enough to be practically assembled into an array.

Array Diameter (D = 10-30 mm): This range was chosen to match the typical scale of a single ‘taxel’ (tactile pixel) in robotic fingertips or e-skin applications. As our analysis in Section 3.4 shows, there is a trade-off between spatial resolution (which is enhanced at smaller diameters) and the practical challenges of assembling smaller, more complex arrays. The 10-30 mm range represents a practical and relevant design space.

Sensing Range (up to 15 mm): This distance was a key parameter of our investigation. A primary goal was to demonstrate that our proposed Halbach array, using strong permanent magnets, could achieve true 'remote' or 'decoupled' sensing. We aimed to verify that, at this significant 15 mm distance, the sensor could still detect a millitesla (mT)-level signal. This is a critical differentiator from many soft magnetic composite sensors, which often only provide microtesla (uT)-level signals at such distances, resulting in a much lower signal-to-noise ratio (SNR). Our simulations confirm the Halbach design achieves this, maintaining a strong signal for high-fidelity sensing.

We have incorporated these detailed explanation into the revised manuscript.

(Main manuscript) “The baseline design parameters were chosen to be representative of tactile sensing applications, balancing performance with practical implementation. The individual magnets (sintered NdFeB N52, 2mm diameter, 2mm height) were selected as they provide the highest remanent flux density (approx. 1.44 T) in a compact, commercially available form factor suitable for embedding in a soft substrate. The array diameter (D) range of 10-30 mm was chosen to match the typical scale of a single tactile pixel for robotic fingertips and e-skin. This range also represents a practical trade-off between spatial resolution, which our analysis in Section 3.4 shows is enhanced at smaller diameters, and the physical challenges of precisely assembling the array. The remote sensing distance (analyzed up to z = -15 mm) was a key parameter of this investigation. A primary objective was to verify that the Halbach array could maintain a strong, millitesla (mT)-level signal at this significant distance. This contrasts with many soft-composite-based sensors, which often drop to the microtesla (uT) level at similar distance, resulting in a much lower signal-to-noise ratio (SNR).”

  1. The manuscript contains several typographical and grammatical errors that need to be corrected for clarity and professionalism.

Response: We sincerely thank the reviewer for this helpful comment. We have carefully proofread the entire manuscript and corrected all typographical, grammatical, and formatting errors to improve clarity, readability, and professionalism. The revised version has also been rechecked using both manual editing and automated grammar review tools to ensure linguistic accuracy and consistency throughout the text. Several corrections are listed below.

[Section 2.1] Original: “This work mainly focused on the magentic field generation section...”

Revised: “This work mainly focused on the magnetic field generation section...”

[Section 3.4] Original: “Same to previous analysis, each magnet was modeled as...”

Revised: “Similar to previous analysis, each magnet was modeled as...”

[Section 5, Conclusions] Original: “...dominant X-Z plane coupling, and monotonic variation with rotation angle and distance features essential for achieving...”

Revised: “...dominant X-Z plane coupling, and monotonic variation with rotation angle and distance, features essential for achieving...”

Reviewer 3 Report

Comments and Suggestions for Authors

Submitted manuscript is devoted to the development of the numerical investigation of Halbach-Array-based flexible magnetic sensors for wide-range deformation detection. Flexible magnetic field sensors for deformation detection were under studies for years and authors must provide objective introduction with short comments related to contributions of different groups. Authors may wish to examine following works or find representative equivalents (Thin-Film Magnetoimpedance Structures onto Flexible Substrates as Deformation Sensors DOI: 10.1109/TMAG.2016.2629513; Static and dynamic study of magnetic properties in FeNi film on flexible substrate, effect of applied stresses DOI: 10.1140/epjb/e2012-30274-0; Flexible, transparent electronics for biomedical applications DOI: 10.1109/ECTC.2013.6575617).

Introduction has rather diffuse structure and does not contain short and clear description at the end of it. There is no need to describe findings in the Introduction, they must be given at the end of the manuscript in the conclusions part.

Work looks like industrial report in which all possible calculations were included. As a result, it contains a lot of figures without proper analysis and some of them (see figures 6, 8) are not even providing readable symbols – collection of non-analyzed data does not add value to the work.

Not a single graph shows experimental errors (see figure 8 H as an example? Where it is strictly necessary).  

Discussion is very general and does not contain valuable arguments. The same for conclusions, they must provide numerical results and future trends should be moved to the discussion section.

Authors must provide comparison with existing theoretical or experimentally available prototypes of flexible magnetic sensors for wide-range deformation detection

Author Response

Response to Reviews

We thank the reviewers for the helpful review. We have addressed all the comments in the revised manuscript and the response letter. The original reviews are included in this response letter, where the reviewers’ comments are in blue. Our responses are in black. The revised parts of the manuscript are included and highlighted in yellow. References are listed at the end of the response letter.

Reviewer #3:

Comments:

Submitted manuscript is devoted to the development of the numerical investigation of Halbach-Array-based flexible magnetic sensors for wide-range deformation detection. Flexible magnetic field sensors for deformation detection were under studies for years and authors must provide objective introduction with short comments related to contributions of different groups. Authors may wish to examine following works or find representative equivalents (Thin-Film Magnetoimpedance Structures onto Flexible Substrates as Deformation Sensors DOI: 10.1109/TMAG.2016.2629513; Static and dynamic study of magnetic properties in FeNi film on flexible substrate, effect of applied stresses DOI: 10.1140/epjb/e2012-30274-0; Flexible, transparent electronics for biomedical applications DOI: 10.1109/ECTC.2013.6575617).

Response: We sincerely thank the reviewer for this constructive feedback. We agree that our original Introduction did not sufficiently cover the diverse field of flexible magnetic sensors and should properly credit the contributions of different groups. To address this, we have revised the Introduction to provide a more objective and comprehensive overview of the field.

We have broadened the scope of the first paragraph to include the general context of flexible electronics for biomedical applications, incorporating the reviewer's suggested citations.

We have revised the third paragraph to explicitly differentiate between the various strategies in magnetic sensing. We now include different sensing modalities, including the magnetoimpedance (MI) and magneto-elastic film approaches suggested by the reviewer, as well as other state-of-the-art works.

This revision better frames the specific strategy we are building upon, the composite-over-Hall-sensor model, as one important branch of this broader field. This allows us to then clearly justify, in the subsequent paragraph, the specific challenges within that branch that our Halbach-array design aims to solve.

We have also condensed the final paragraph to be a short and clear description of the paper’s objective, removing any discussion of the findings, which are now reserved for the Conclusions. We believe these revisions make the Introduction more comprehensive and more clearly focused.

(Main manuscript) “To overcome these limitations, magnetic-based tactile sensing has emerged as a promising approach due to its unique combination of mechanical flexibility, environmental robustness, and high spatial resolution [33-36]. This is a broad field encompassing diverse strategies. Some approaches utilized the magneto-elastic effect, where applied stress directly modulates the magnetic properties of a flexible film (e.g., FeNi) [37]. Other works have developed flexible sensors based on magnetoimpedance (MI) structures that are highly sensitive to deformation [38]. A third, widely-adopted strategy, which forms the basis for our work, involves a soft composite layer (a soft matrix embedded with micro- or nano-magnets) that mechanically deforms over a fixed Hall or magnetoresistive sensing element. In this configuration, deformation induces displacement or rotation of the magnetic source, causing measurable magnetic field variations [39-41]. This composite-based Hall-effect approach, in particular, offers several key advantages: (1) Easy deployment and structural simplicity, since magnetic coupling allows non-contact signal transmission without complex wiring or optical paths; (2) Magnetic-electrical decoupling, which enables the sensitive electronics (e.g., Hall elements) to be isolated from mechanical strain, thereby improving device durability and long-term stability; (3) High spatial resolution, achievable by designing magnetic arrays or encoding field gradients for super-resolution tactile mapping; and (4) Compatibility with soft materials, enabling conformal integration onto curved or deformable surfaces [42-44].

Despite these advantages, magnetic tactile sensors still face several key challenges that limit their performance in practical applications. First, as noted in designs by Yan et al. [45], when the Hall sensor is directly attached behind the magnetic film, the rigid-soft interface restricts the mechanical deformability, leading to nonlinear response under large strain. To address this, “split-type” or decoupled architectures have been proposed, such as by Dai et al. [33] and Sundaram et al. [46], which physically separate the deformable magnetic layer from the rigid sensor. These designs improve robustness and enable remote sensing. Second, the magnetic field from conventional magnets is non-uniform and three-dimensional, complicating calibration. While soft magnetic composites allow for novel field profiling (e.g., sinusoidal or centripetal) to achieve force decoupling, the resulting 3D fields often require complex analytical models or machine learning for interpretation. Third, the magnetic signal intensity rapidly decays with distance, which constrains the sensing range and makes it difficult to achieve accurate measurements when the magnetic and electronic units are separated. These issues hinder the realization of high-precision, wide-range magnetic tactile sensors that can operate reliably under large deformation, an essential requirement for wearable and robotic E-skins [36,45,47-49].

To tackle these challenges, we propose a Halbach-array-based magnetic tactile sensor that fundamentally restructures the spatial relationship between the magnetic source and the sensing element. In our design, a soft PDMS-based magnetic layer embedded with k = 2 Halbach-configured magnetic cubes are structurally decoupled from a rigid Hall sensing unit placed at a fixed remote location. Here, we present a comprehensive numerical investigation using COMSOL Multiphysics to establish a theoretical framework for this sensor design. This study systematically analyzes the magnetic field distribution, uniformity and strength as a function of key design parameters, to provide a foundational guide for the development of wide-range, high-precision flexible magnetic sensors.”

  1. 1. Introduction has rather diffuse structure and does not contain short and clear description at the end of it. There is no need to describe findings in the Introduction, they must be given at the end of the manuscript in the conclusions part.

Response: We sincerely thank the reviewer for this valuable feedback on the structure and clarity of the Introduction. We realized that the final paragraphs were discussing results and implications that are better suited for the Conclusions section. Moreover, we have carefully polished the Introduction and the whole manuscript, and all the changes have been marked in Yellow in the revised manuscript.

(Main manuscript, end of Introduction) “Despite these advantages, magnetic tactile sensors still face several key challenges that limit their performance in practical applications. First, as noted in designs by Yan et al. [45], when the Hall sensor is directly attached behind the magnetic film, the rigid-soft interface restricts the mechanical deformability, leading to nonlinear response under large strain. To address this, “split-type” or decoupled architectures have been proposed, such as by Dai et al. [33] and Sundaram et al. [46], which physically separate the deformable magnetic layer from the rigid sensor. These designs improve robustness and enable remote sensing. Second, the magnetic field from conventional magnets is non-uniform and three-dimensional, complicating calibration. While soft magnetic composites allow for novel field profiling (e.g., sinusoidal or centripetal) to achieve force decoupling, the resulting 3D fields often require complex analytical models or machine learning for interpretation. Third, the magnetic signal intensity rapidly decays with distance, which constrains the sensing range and makes it difficult to achieve accurate measurements when the magnetic and electronic units are separated. These issues hinder the realization of high-precision, wide-range magnetic tactile sensors that can operate reliably under large deformation, an essential requirement for wearable and robotic E-skins [36,45,47-49].

To tackle these challenges, we propose a Halbach-array-based magnetic tactile sensor that fundamentally restructures the spatial relationship between the magnetic source and the sensing element. In our design, a soft PDMS-based magnetic layer embedded with k = 2 Halbach-configured magnetic cubes are structurally decoupled from a rigid Hall sensing unit placed at a fixed remote location. Here, we present a comprehensive numerical investigation using COMSOL Multiphysics to establish a theoretical framework for this sensor design. This study systematically analyzes the magnetic field distribution, uniformity and strength as a function of key design parameters, to provide a foundational guide for the development of wide-range, high-precision flexible magnetic sensors.”

  1. 2. Work looks like industrial report in which all possible calculations were included. As a result, it contains a lot of figures without proper analysis and some of them (see figures 6, 8) are not even providing readable symbols – collection of non-analyzed data does not add value to the work.

Response: We sincerely thank the reviewer for this critical and very helpful feedback. We realized that the manuscript, in its original form, presented a large volume of simulation data without sufficient scientific analysis of its implications for sensor design. To address these significant issues, we have made the following major revisions:

  1. Figure legibility and renumbering: We have re-rendered Figure 7 with significantly larger fonts and clearer legends to ensure all data is legible. We also identified a significant error in our original manuscript: there were two figures labeled “Figure 6”. We have corrected this. The original first “Figure 6” (z-axis profile) remains Figure 6. The original second “Figure 6” (the large grid) is now correctly labeled Figure 7. The original “Figure 7” is now Figure 8, and the original “Figure 8” (diameter comparison) is now Figure 9.
  2. Adding in-depth analysis: We have substantially rewritten the corresponding text in Sections 3.3 and 3.4 to move from simple description to in-depth analysis. We no longer just state what the plots show (e.g., “the profile is symmetric”) but now analyze why these results are important for a functional sensor.

- For the new Figure 7: Our revised text now explicitly analyzes what the different field components tell us. We discuss how the near-identical Bx and Btotal plots prove the quasi-1D nature of the field; how the linearity of Bx with rotation forms the primary sensing mechanism; how the negligible By component confirms minimal crosstalk; and, most importantly, how the non-linear profile of Bz near the magnets could serve as a unique spatial “fingerprint” to simultaneously sense both rotation and axial distance (z), a key insight that was previously missing.

- For the new Figure 9: We have expanded the analysis to focus on the design trade-offs revealed by the data. We now explicitly state that the “pronounced spatial variation” in the 10 mm array translates to higher sensitivity and spatial resolution (dB/dx). We also clarify why the 30 mm array’s non-monotonic profile is “undesirable”, as it introduces signal ambiguity (one Bx value could correspond to two different positions). This analysis now provides a clear, data-backed justification for our design choices.

We believe these revisions add significant scientific value, directly connecting the raw simulation data to the sensor's design principles and practical performance.

(Main manuscript, section 3.3) “We further analyzed the magnetic flux density distribution along the X-axis at different Z-axis heights within the Halbach array, including the total magnetic flux density (B) and its orthogonal components (Bx, By, and Bz), as shown in Figure 7. The first and second columns of Figure 7 provide a comparison of the total magnetic flux density (B) and the X-directional component (Bx), which reveals that their profiles and magnitudes are nearly identical. This is a crucial finding, as it quantitatively confirms the success of the Halbach array’s dimensionality reduction. The By and Bz components are so effectively suppressed that the total field is almost entirely dominated by Bx. Furthermore, the plots show a clear, predictable, and highly linear variation in Bx magnitude in response to the rotation angle (θ). This linear Bx - θ relationship forms the primary sensing principle for a simple rotational/shear sensor.

The Z-directional component in the third column shows a complex, non-linear profile. Near the magnets (e.g., z=0 mm and z=-5 mm), it exhibits a sharp polarity reversal at X=0 mm. As the distance Z increases, this non-linear profile evolves into a simpler, quasi-linear one. This non-linearity is not simply an error, it provides a unique spatial “fingerprint”. A sensor placed close to the array (z=-5 mm) could potentially use a machine learning model to read the non-linear relationship between Bx and Bz to simultaneously decouple and sense both rotation and axial distance. A sensor placed further away (z=-15 mm) would lose this z-position information but would benefit from a simpler, more linear response.

The Y-directional component (fourth column) shows only weak, μT-level high-frequency oscillations, which are negligible. This is a highly desirable result, as it confirms that crosstalk from side-to-side (y-axis) motion or misalignment is minimal, greatly simplifying the sensor’s calibration and operation.

A key claim of this work is that the Halbach configuration enables “high sensitivity” sensing. We must clarify this claim in the context of Signal-to-Noise Ratio (SNR). As quantified in our analysis (Figure 6A), the use of permanent magnets in a Halbach array generates a strong, millitesla (mT)-level signal, even at a remote 15 mm distance. This signal is 2-3 orders of magnitude stronger than the μT-level signals typically reported by sensors using soft magnetic composites at similar distances [33,48]. This high-strength signal is a critical advantage for practical applications, as it provides a high SNR, enabling robust and high-fidelity measurements even in noisy environments. The sensor's response is then clearly and linearly resolved, as shown in Figure 6B.”

(Main manuscript, section 3.4) “Figure 9A-C present the Bx distributions in the x–y plane at Z = –6 mm within the array. This analysis reveals a critical design trade-off. The smaller-diameter array (10 mm) generates stronger fields and, more importantly, more pronounced spatial variation (i.e., a steeper dBx/dx gradient). This implies that a smaller array diameter provides higher spatial resolution and greater sensitivity to positional changes. Conversely, the 30 mm array exhibits a non-monotonic, “M-shaped” profile, with two distinct edge peaks. This is highly undesirable for sensing, as a single Bx value could correspond to two different x-positions, creating signal ambiguity.

Figure 9D-H shows the magnetic field distributions along the Z-axis. While the overall field magnitude decreases with increasing array diameter, the fundamental spatial distribution pattern and linear response to rotation remain consistent. This result is significant as it demonstrates the generality and scalability of the Halbach design principle.

Based on these findings, minimizing the array diameter (e.g., 10 mm or 20 mm) is recommended to maximize spatial resolution and sensitivity while avoiding the signal ambiguity observed in larger arrays.”

  1. 3. Not a single graph shows experimental errors (see figure 8 H as an example? Where it is strictly necessary).

Response: We thank the reviewer for this comment. We acknowledge that experimental data must be presented with error bars to be statistically meaningful.

However, we must clarify that this entire manuscript is a purely numerical investigation performed using COMSOL Multiphysics, as stated in the Abstract and Section 2.2. The plots in the manuscript, including Figure 8 (now renumbered to Figure 9), show deterministic simulation data, not experimental measurements. As the results are from a finite element model, there are no experimental stochastic errors (e.g., measurement noise, fabrication tolerances) to plot as error bars. The plots represent the direct, repeatable output of the simulation.

We recognize that our presentation made the data appear experimental. We have revised the text in Section 3.4 and the caption for Figure 9 (formerly Figure 8) to more explicitly state that this is simulated data. We also acknowledge in the “Conclusions” section that a full experimental validation, which would include an analysis of experimental errors, is a critical next step for this work.

(Main manuscript, Section 3.4) “To systematically investigate the influence of array diameter on magnetic field modulation, a series of simulations was conducted.”

(Main manuscript, Caption for Figure 9) “Figure 9. Simulated magnetic field characteristics of Halbach arrays with different diameters (10 mm, 20 mm, 30 mm). (A–C) Simulated magnetic field distributions of the Bx component in the x–y plane at Z = –6 mm, y = 0 mm. (D–F) Simulated magnetic field distributions along the Z direction, ranging from 5mm to -20 mm. (G–H) Quantitative analysis from the simulation data, showing the Z-directional magnetic field peak intensity and position.”

  1. 4. Discussion is very general and does not contain valuable arguments. The same for conclusions, they must provide numerical results and future trends should be moved to the discussion section.

Response: We sincerely thank the reviewer for this valuable feedback. We agree that our original Discussion was too general and that the Conclusions section was a mix of findings and future work. To address these concerns, we have made a major revision of the final two sections of the manuscript:

Revised Discussion (Section 4): We have moved all discussions of “future trends” and “limitations” from the Conclusions to this section, as suggested.

To add “valuable arguments”, we have incorporated the new comparative analysis (Table 1), which contrasts our approach with prior art, highlighting the key advantage in signal strength (mT vs. μT). We also added a new, detailed discussion on the practical challenges of experimental validation and a proposed workflow for real-world implementation. This directly connects our theoretical work to a practical application.

We believe these substantial revisions create a much stronger manuscript. The Discussion section now provides a robust analysis of our work’s context and practical application, while the Conclusions section serves as a clear summary of what this numerical study has achieved.

(Main manuscript, Discussion section) “Importantly, these results indicate that optimizing the array diameter can significantly enhance spatial resolution and SNR without compromising the field’s structural integrity. The findings also suggest that deep learning-based models could exploit the nonlinear spatial correlations observed in Bx and Bz to achieve higher accuracy in multi-axis force or deformation sensing, paving the way for adaptive calibration strategies in soft magnetic sensor systems. To clearly situate our work, Table 1 provides a comparative analysis against other state-of-the-art magnetic sensing methods.

Table 1. Comparative analysis of representative magnetic tactile sensing methods.

Method

Magnetic Source

Field Configuration

Sensing Principle

Signal Strength (at >10mm)

Hellebrekers (2019) [37]

Soft composite (particles)

Random particle distribution

Single 3-axis Magnetometer

μT-level (inferred)

Yan (2021) [42]

Soft composite (film)

Sinusoidal magnetization

3-axis Hall Sensor

μT-level

Becker (2022) [46]

(External magnet)

3D micro-origami cube

Anisotropic Magnetoresistance (AMR)

(N/A)

Rehan (2022) [33]

Discrete permanent magnets

4-magnet simple array

4x single-axis Hall Sensors

≈7.3 mT @ 6mm

Dai (2024) [32]

Soft composite (film)

Centripetal magnetization

3-axis Hall Sensor

≈40μT @ 20mm

Our Proposed Work

Discrete permanent magnets (N52)

Halbach Array

3-axis Hall Sensor

≈1mT @ 15mm

Besides, experimental validation represents the next critical step. We are currently fabricating and testing physical prototypes to confirm the predicted field structures and evaluate sensing performance. The practical implementation will require precise magnet alignment and mechanical stabilization during fabrication. Furthermore, as real deformation induces complex six degree-of-freedom motion, we plan to adopt a data-driven calibration strategy combining multi-axis force measurements with machine learning to map the nonlinear relationship between magnetic signals and external stimuli. Leveraging the high SNR and structured field distribution of the Halbach design, these efforts aim to enable robust, wide-range, and high-precision tactile sensing for soft robotics and intelligent systems.

While this numerical study establishes the theoretical foundation for a Halbach-array-based magnetic tactile sensor, several limitations remain. The present model isolates in-plane magnet rotation to clarify its role in field modulation, but it does not yet account for coupled deformation modes, such as translational and out-plane displacements, or the nonlinear mechanical behavior of the elastomer substrate. These factors will inevitably influence the real sensor’s magnetic response and will be incorporated in future coupled magneto-mechanical simulations.”

(Main manuscript, Conclusion section) “This study presents a comprehensive numerical investigation of a Halbach-array-based magnetic tactile sensor, demonstrating a design that enables directional magnetic field modulation and remote, decoupled sensing. The primary findings of this simulation study are:

  1. High Signal Strength: The k=2 Halbach configuration, combined with N52 permanent magnets, generates a high-strength, millitesla (mT)-level magnetic field, enabling robust remote detection at distance up to 15 mm.
  2. Field Dimensionality Reduction: The array successfully tailors the magnetic field, reducing its spatial dimensionality from a complex 3D profile to a quasi-1D distribution, as quantified in Table 1. The transverse components (By and Bz) are suppressed by nearly two orders of magnitude compared to Bx near the central axis.
  3. Linear & Stable Response: The sensor’s response to magnet rotation, a proxy for film deformation, is highly predictable. As shown in Figures 4-7, the field distribution remains spatially stable, with the rotation primarily modulating the field’s magnitude, not its direction. This response, measured by the peak Bx amplitude and position, exhibits excellent linearity (R2 > 0.99).
  4. Design Guideline: The parameter study (Figure 9) established a clear design trade-off: an array diameter of 10-20 mm provides the optimal balance, maximizing spatial resolution while avoiding the signal ambiguity seen in larger arrays.

Taken together, these findings establish a robust theoretical framework for this sensor. The intrinsic 1D field orientation and high SNR simplify calibration and form a solid physical foundation for the future development of high-precision, wide-range flexible magnetic sensors for robotics and wearable applications.”

  1. 5. Authors must provide comparison with existing theoretical or experimentally available prototypes of flexible magnetic sensors for wide-range deformation detection.

Response: We thank the reviewer for this important suggestion. We agree that a direct comparison to existing prototypes is essential to highlight the contribution of our numerical work. We have addressed this comment by adding a new comparative analysis (Table 1) in the Discussion section, as in the response to your previous question.

This table explicitly compares our proposed theoretical sensor to several key experimentally available prototypes from recent literature. We believe this new table and analysis directly address the reviewer’s comment by providing the requested comparison and strengthening the paper's contribution.

References

  1. Dai, H.; Zhang, C.; Pan, C.; Hu, H.; Ji, K.; Sun, H.; Lyu, C.; Tang, D.; Li, T.; Fu, J. Split‐type magnetic soft tactile sensor with 3D force decoupling. Advanced Materials 2024, 36, 2310145.
  2. Lin, X.; Han, M. Recent progress in soft electronics and robotics based on magnetic nanomaterials. Soft Sci 2023, 3, 14.

Round 2

Reviewer 1 Report

Comments and Suggestions for Authors

Thank you for implementing these changes! I think it conveys the work that was done in a clearer way. One comment is to cite this claim made in 2.2: "The array diameter (D) range of 10-30 mm was chosen to match the typical scale of a single tactile pixel for robotic fingertips and e-skin. This range also represents a practical trade-off between spatial resolution, which our analysis in Section 3.4 shows flux density is enhanced at smaller diameters, and the physical challenges of precisely assembling the array."

Author Response

Comments:

Thank you for implementing these changes! I think it conveys the work that was done in a clearer way.

Response: We thank the reviewer for the helpful suggestions so that we can improve the manuscript to its current form. We also carefully addressed the new comments, while the detailed, point-by-point responses to each comment are presented below.

1) One comment is to cite this claim made in 2.2: “The array diameter (D) range of 10-30 mm was chosen to match the typical scale of a single tactile pixel for robotic fingertips and e-skin. This range also represents a practical trade-off between spatial resolution, which our analysis in Section 3.4 shows flux density is enhanced at smaller diameters, and the physical challenges of precisely assembling the array.”

Response: We thank the reviewer for the suggestion. We have now revised this sentence in Section 2.2 to include citations from several key experimental works on tactile sensors [] that use single tactile pixels or sensing elements in the 8mm to 20mm range, which validates our chosen parameter space.

(Main manuscript, Section 2.2) “The array diameter (D) range of 10-30 mm was chosen to match the typical scale of a single tactile pixel for robotic fingertips and e-skin, which often required magnetic unit ranging from approximately 8 mm to 20 mm in diameter or width [35,36,47].”

Reviewer 3 Report

Comments and Suggestions for Authors

Submitted revised version of the manuscript describes the development of the numerical investigation of Halbach-Array-based flexible magnetic sensors for wide-range deformation detection. Flexible magnetic field sensors for deformation detection were under studies for years and authors made reasonable improvements in the description of over the world made contributions. However, the goal of the study must be given at the end of the Introduction as short paragraph and preferably in in an impersonal form.

Work is better now but it still looks like industrial report in which all possible calculations were included. As a result, it contains a lot of figures without proper analysis. The number of graphs must be reduced because at the present state they are rather collecting the data but not providing the evidence of the analysis. Figures 5? 7? 9 must be shortened and optimized.

Not a single graph shows experimental errors (see figure 8 H as an example? Where it is strictly necessary). For numerical studies authors use real parameters, which can be defined with certain accuracy as well as all solutions usually have degree of instability. These points must be discussed.

Conclusions cannot be written as a list of phrases. The text should be comprehensive, self-consistent, reflect the main contribution, and include numerical data.

Author Response

Comments:

Submitted revised version of the manuscript describes the development of the numerical investigation of Halbach-Array-based flexible magnetic sensors for wide-range deformation detection. Flexible magnetic field sensors for deformation detection were under studies for years and authors made reasonable improvements in the description of over the world made contributions. However, the goal of the study must be given at the end of the Introduction as short paragraph and preferably in an impersonal form.

Response: We sincerely thank the reviewer for acknowledging the improvements made in our revised manuscript. We also appreciate the valuable guidance regarding the final paragraph of the Introduction. We have carefully revised the Introduction to ensure its concise and in an impersonal form, as suggested. The revised version now focuses strictly on the study’s objective and the proposed sensor architecture.

(Main manuscript, Introduction section) “To address these challenges, this study proposes a Halbach-array-based magnetic tactile sensor in which a soft PDMS magnetic layer, embedded with k = 2 Halbach-configured magnetic cubes, is structurally decoupled from a rigid, remotely positioned Hall sensing unit. A comprehensive numerical framework is developed using COMSOL Multiphysics to investigate the magnetic field distribution, uniformity, and strength as functions of key design parameters. The aim is to establish design principles that enable wide-range, high-precision, and mechanically flexible magnetic sensing.”

  1. Work is better now but it still looks like industrial report in which all possible calculations were included. As a result, it contains a lot of figures without proper analysis. The number of graphs must be reduced because at the present state they are rather collecting the data but not providing the evidence of the analysis. Figures 5? 7? 9 must be shortened and optimized.

Response: We thank the reviewer for this helpful comment. We agree that some of the figures in previous version is too large and it’s hard to grasp key information at first sight. To tackle this problem, we have now shortened Figure 5 and 7, and more description and analysis has been added in the revised manuscript, while a new table has been added to quantify the magnetic field strength difference at different direction in magnitude. Moreover, we have condensed the description and text length through out the whole manuscript, while all the changes have been marked in Yellow.

(Main manuscript, Section, 3.2) “Figure 5 shows the simulated magnetic flux density in horizontal cross-sections of the Halbach array at different probe heights (z) and rotation angles (θ). Rows correspond to axial positions (z = 5 mm, −5 mm, −10 mm, −15 mm) and columns to rotation angles (θ = 0°–30°). Because the sensor is positioned below the array (negative z), the top row (z = 5 mm) represents the suppressed field side, while the lower rows illustrate the enhanced sensing region.

The simulations reveal a clear rotation-induced asymmetry. At θ = 0°, the field is symmetric about z = 0, with identical distributions at z = +5 mm and −5 mm. Once rotation begins, this symmetry is lost: the Halbach configuration strengthens the field in the sensing region (z < 0) and weakens it above the array (z > 0). With increasing θ, the field evolves nonlinearly despite the axial peak flux varying nearly linearly, highlighting spatial complexity central to the sensor’s modulation behavior.

For a fixed probe height, increasing θ causes the field pattern to rotate synchronously, with high- and low-field regions shifting around the central axis. This strong coupling between magnet orientation and field direction allows the global field both to be tuned and to encode the array’s rotational state.

Overall, probe height primarily affects the magnitude and uniformity of the flux density, while rotation angle sets its directional features. The consistent field pattern during rotation suggests robust immunity to external magnetic disturbances.

Figure 5. Simulated magnetic flux density distribution in horizontal cross-sections of the Halbach array at varied probe heights (z = 5, –5, –10, –15 mm) and rotation angles (θ = 0°–30°). The results show that smaller |z| yields higher flux density and steeper gradients, while rotation (θ) synchronously modulates the field orientation and distribution pattern, demonstrating the sensing ability of applied force.”

(Main manuscript, Section 3.3) “Figure 7 presents the magnetic flux density along the X-axis at several Z-heights within the Halbach array. Across all positions, the field is dominated by the Bx component and By and Bz remain strongly suppressed, while By showing only μT-level oscillations (Table 1). As the difference in these two directions is on the order of 104-107, this confirms excellent rejection of lateral (y-axis) disturbances and minimal crosstalk due to misalignment.

The Bx profiles show smooth, predictable, and nearly linear changes with rotation angle θ, forming a clean basis for rotation or shear sensing. As Z decreases toward the magnets, the curves become sharper and exhibit a larger dynamic range, while at greater distances (e.g., z = −15 mm) the profiles flatten but maintain strong linearity.

The Bz component exhibits more complex behavior. Close to the magnets (e.g., z = −3 mm), B? shows a pronounced polarity flip at x = 0 and a distinctly nonlinear shape. With increasing distance, this pattern gradually transitions into a more monotonic, quasi-linear form. This evolution means that Bz encodes additional spatial information: at small Z, the nonlinear signature could allow simultaneous estimation of rotation and axial position using data-driven models, whereas at large Z the response becomes simpler but loses Z-specific detail. Results in Figure 7 demonstrate that Bx provides a clean and highly linear rotational signal, Bz carries depth-dependent spatial cues, and By remains negligible, together offering a robust magnetic field environment for stable and decoupled sensing.

A key claim of this work is the high sensitivity enabled by the Halbach configuration, expressed through its strong SNR. As shown in Figure 6A, the array produces mT-level signals even at 15 mm, 2-3 orders stronger than the μT-level fields typical of soft magnetic composites [35,50]. This large signal ensures a high SNR and a clean, linear response, as seen in Figure 6B.”

Figure 7. Magnetic flux density and its directional components along the X-axis at various Z-axis heights within the Halbach array. Each column corresponds to a field component (Bx, By, and Bz), and each row represents a different Z position. The curves depict magnetic flux density distributions along the X-axis from –8 mm to 8 mm at multiple rotation angles. The inset images provide an enlarged view of the central region (–3 mm to 3 mm).

Table 1. Average magnetic flux densities and signal dominance ratios.

Distance

Avg. Bx (mT)

Avg. By (mT)

Order of (Bx/By)

-3 mm

-9 mm

-15 mm

* X-coordinate is ranging from -5 to 5 mm.

  1. 2. Not a single graph shows experimental errors (see figure 8 H as an example? Where it is strictly necessary). For numerical studies authors use real parameters, which can be defined with certain accuracy as well as all solutions usually have degree of instability. These points must be discussed.

Response: We thank the reviewer for the valuable clarification. We now recognize that the concern relates not to experimental error but to the numerical stability of the COMSOL model and the sensitivity of the results to uncertainties in the input parameters. We have conducted additional analyses to address both points.

Numerical stability. To evaluate the “degree of instability” of the numerical solution in our simulation process, we performed a mesh-convergence study. The mesh was refined until the change in peak magnetic flux density fell below 0.1%. This confirms that the results are numerically stable and independent of mesh discretization. A corresponding statement has been added to Section 2.2 to make it clearer.

Parameter sensitivity. To examine the “certain accuracy” of the material parameters, we carried out a sensitivity analysis on the remanent flux density (Bᵣ) of the permanent magnets, the dominant input variable. Given the typical ±2-3% manufacturing tolerance of N52 magnets, we re-ran the simulations for Bᵣ variations of ±3% (from 1.397T to 1.483T). The resulting uncertainty bands have been added to Figure 9G. The analysis shows that although absolute field strength shifts proportionally, the linearity and overall sensing behavior remain unchanged, as well at the position of flux density peak in Z-axis (Figure 9H). This demonstrates that the proposed design is robust to realistic parameter variations. The corresponding discussion has been added to Section 3.4.

(Main manuscript, Section 2.2) “A fine mesh was applied to ensure numerical accuracy, with element sizes of 0.2 mm in the central region, 0.3 mm in the magnets, and 0.32-2.56 mm in the surrounding air. Free tetrahedral elements were used throughout. To ensure the “degree of instability” of the solution was minimized, a mesh-convergence study was performed; the mesh was refined until the change in peak magnetic flux density was below 0.1%, confirming numerical stability and independence from mesh discretization. To isolate the magnetic response induced purely by magnet rotation, the magnets were constrained to rotate in-plane about their central axes (θ up to 60°) without translation. This assumption reflects the dominant deformation mode in the soft magnetic film, where external loading primarily induces small-angle rotations rather than significant translational motion.

While this model captures the essential rotation-driven modulation of the magnetic field, the main mechanism governing sensor sensitivity, it does not include out-of-plane shifts or magnet-matrix interactions that may arise in real deformation. These coupled magneto-mechanical effects will be incorporated in future studies. Figure 2C shows the simulated x-z plane field distribution as θ increases from 0° to 30°. The field remains strongly aligned along the x-axis with only a minor z-component, and its orientation remains stable during rotation, confirming the directional robustness of the Halbach configuration.”

(Main manuscript, Section 3.4) “Furthermore, a parameter uncertainty analysis was applied to the 10 mm array, as shown in Figure 9G. The shaded bands (representing a ±3% variation in remanence, Br) indicate that while the absolute peak intensity scales with magnet strength, the peak position and the highly linear response to rotation remain stable, confirming the robustness of the design.

Based on these findings, minimizing the array diameter (e.g., 10 mm or 20 mm) is recommended to maximize spatial resolution and sensitivity while avoiding the signal ambiguity observed in larger arrays.”

Figure 9. Simulated magnetic field characteristics of Halbach arrays with different diameters (10 mm, 20 mm, 30 mm). (A–C) Simulated magnetic field distributions of the Bx component in the x–y plane at Z = –6 mm, y = 0 mm. (DF) Simulated magnetic field distributions along the Z direction, ranging from 5mm to -20 mm. (GH) Quantitative analysis from the simulation data, showing the Z-directional magnetic field peak intensity and position.

  1. 3. Conclusions cannot be written as a list of phrases. The text should be comprehensive, self-consistent, reflect the main contribution, and include numerical data.

Response: We thank the reviewer for this helpful comment. We have now rewritten the conclusion section to better summarize the whole study in proper form.

(Main manuscript, Conclusion section) “This study presents a comprehensive numerical investigation of a Halbach-array-based magnetic tactile sensor, establishing a theoretical framework for a de-sign that enables directional magnetic field modulation and remote, decoupled sensing. The simulations demonstrate that the k=2 Halbach configuration, combined with N52 permanent magnets, generates a high-strength magnetic field exceeding 1 mT at a remote sensing distance of 15 mm. Crucially, the array successfully tailors the magnetic field, reducing its spatial dimensionality from a complex 3D profile to a quasi-1D distribution, where the transverse components (By and Bz) are suppressed by more than two orders of magnitude relative to the dominant Bx component near the central axis.

Furthermore, the sensor demonstrates a highly stable and predictable response to magnet rotation. The field distribution remains spatially consistent, with the rotation primarily modulating the field’s magnitude rather than its direction. This response, measured by the peak Bx amplitude and position, exhibits excellent linearity (R2 > 0.99) as a function of rotation angle. Finally, the parameter study establishes a clear design guideline, indicating that an array diameter of 10-20 mm provides the optimal balance by maximizing spatial resolution while avoiding the signal ambiguity observed in larger arrays. Collectively, these quantitative findings confirm that the Halbach-array-based design offers a robust physical foundation for high-precision, wide-range, and remote tactile sensing applications.”
